# Genomic analysis of *Plasmodium vivax* field isolates circulating in sub-Saharan Africa
Isabelle Bouyssou[1,2,3] ✉, Lemu Golassa[4], Inès Vigan-Womas[5], Matthieu Schoenhals[6], Arsène Ratsimbasoa[7], Ali Ould Mohamed Salem Boukhary [8], Maria de Fátima Ferreira-da-Cruz[9], Sandrine Houzé[10], Laurence Ma[11], Feng Lu[12], Chetan Chitnis[3], Pascal Campagne [13] ✉ & Didier Ménard [1,3,14,15,16] ✉

*Plasmodium vivax* malaria is a major public health problem outside sub-Saharan Africa. However, an increasing number of *P. vivax* infections in Duffy-negative individuals has been reported across Africa in recent years, raising concerns that the parasites may have evolved alternative pathways to invade reticulocyte and overcome Duffy-negativity. Here, we investigated the global genetic structure and diversity of sub-Saharan African *P. vivax* populations, exploring possible molecular signatures of adaptation to Duffy-negative hosts. We analyzed 204 previously published *P. vivax* genome sequences from Africa, Southeast Asia, the Pacific Coral Triangle, and South America and generated whole-genome sequences of 133 *P. vivax* field isolates collected from 10 sub-Saharan African countries. Our analysis revealed four distinct geographic clusters, with clear contrasts between East/West Africa and the Indian Ocean populations. Despite the limited number of interpretable sequences from Duffy-negative individuals - attributable to low parasitemia - and the lack of clear evidence of selective pressure acting on invasion-related genes of the *P. vivax* parasite populations circulating in sub-Saharan Africa, our study offers valuable insights into the genetic diversity of *P. vivax* and lays the groundwork for future research exploring parasite adaptation to Duffy-negative hosts.

*Plasmodium vivax* malaria remains a major public health concern in South America, Southeast Asia, the Middle East, the Pacific Coral Triangle, and Eastern and Southern Africa[1]. Historically, the predominance of Duffy-negative human populations in sub-Saharan Africa has led to the belief that *P. vivax* transmission is limited in this region[2]. This paradigm stems from evidence that *P. vivax* merozoites rely on the interaction between the Duffy Antigen Receptor for Chemokines (DARC) and the P. vivax Duffy Binding Protein (PvDBP) to invade erythrocytes[3]. However, recent studies have reported cases of *P. vivax* infections in Duffy-negative individuals across Africa[4–14], challenging the conventional understanding and raising questions about potential alternative invasion pathways.

Despite these intriguing observations, robust genomic data from Duffy-negative individuals remain limited due to the technical challenges of obtaining high-quality sequences from low-parasitemia infections. Consequently, the molecular mechanisms underlying *P. vivax* invasion into Duffy-negative reticulocytes remain poorly understood. Previous studies have described duplications of the *pvdbp* gene[15,16] in regions such as

Madagascar, where Duffy-positive and Duffy-negative populations coexist, and Southeast Asia[17,18]. Additional work has identified homologs such as the P. vivax Erythrocyte Binding Protein (PvEBP/PvDBP2)[19], which preferentially binds immature (CD71[high]) reticulocytes in Duffy-positive individuals but shows minimal binding to Duffy-negative reticulocytes[20]. In parallel, other ligands like PvRBP2a and PvRBP2b have been identified as key players in reticulocyte recognition prior to invasion[21,22]. However, no conclusive evidence has demonstrated that these genes facilitate Duffy-independent invasion.

Here, we focus on analyzing the global genetic diversity and population structure of *P. vivax* across Africa and other endemic regions to better understand the selective pressures shaping parasite populations. We provide whole-genome sequences of 133 *P. vivax* field isolates from 10 African countries (Angola, Burundi, Comoros, Djibouti, Egypt, Eritrea, Ethiopia, Madagascar, Mauritania, and Sudan) using selective whole-genome amplification (sWGA) and next-generation sequencing. Our dataset was enriched with 204 publicly available sequences from Africa, Southeast Asia,

A full list of affiliations appears at the end of the paper. ✉e-mail: isabelle.bouyssou@outlook.com; pascal.campagne@pasteur.fr; dmenard@pasteur.fr; dmenard@unistra.fr

the Pacific Coral Triangle, and South America[23], for a total of 337 samples across 18 countries.

To ensure a robust analysis, we assessed within-host diversity by calculating $F_{ws}$ values to distinguish monoclonal infections from polyclonal infections and to estimate heterozygosity. Our study primarily explores the global genetic structure of *P. vivax*, identifies regions under selection, and examines population dynamics in areas with a high prevalence of Duffy-negativity. While the limited availability of high-quality Duffy-negative sequences prevents definitive conclusions on parasite adaptation, this work lays the foundation for future studies with larger and more balanced datasets.

## Results

### Samples

A total of 133 *P. vivax* field isolates were obtained between 2016 and 2021 from symptomatic patients originating from ten sub-Saharan African countries. Samples were collected locally at health centers in Ethiopia ($N = 66$), Madagascar ($N = 27$), Mauritania ($N = 2$), and Angola ($N = 12$). Additional DNA extracts were obtained from symptomatic travelers returning to France from the Comoros ($N = 8$), Mauritania ($N = 5$), Djibouti ($N = 3$), Sudan ($N = 3$), Madagascar ($N = 2$), Eritrea ($N = 2$), Ethiopia ($N = 1$), Burundi ($N = 1$), and Egypt ($N = 1$).

We amplified *P. vivax* DNA using sWGA to address the challenge of low parasitemia in clinical samples and successfully sequenced 72/133 *P. vivax* genomes using high-throughput sequencing. Among the successfully sequenced genomes, 18/72 (25%) were from homozygous Duffy-positive patients, 52/72 (72.3%) were from heterozygous Duffy-positive patients, and 2/72 (2.7%) had undetermined Duffy status. Despite our efforts, *P. vivax* genome sequences from 18 homozygous Duffy-negative patients could not be properly exploited due to insufficient parasitemia levels, which resulted in low DNA amplification and inadequate sequencing quality (see "Methods" for details). This limitation highlights the ongoing technical challenges in obtaining high-quality genomic data from Duffy-negative individuals, an issue that remains a bottleneck for understanding potential adaptations of *P. vivax* to these hosts.

To expand the scope of our analysis, we enriched our dataset with 204 publicly available *P. vivax* genome sequences from prior studies. These included sequences from Southeast Asia (Cambodia, Thailand, Vietnam), the Pacific Coral Triangle (Indonesia, Malaysia, Papua New Guinea), South America (Brazil, Colombia), and Africa (Ethiopia, Madagascar, Mauritania). Full details, including sample metadata and references, are provided in Table 1 and Supplementary Information (Table S1).

### Genomic data

We aligned the sequencing reads to the PvP01 (v.48) reference genome using bwa mem and used GATK 4.0 (Genome Analysis Toolkit) following best practices guidelines to identify Single Nucleotide Polymorphisms (SNPs). Genotypes were defined in diploid mode. To ensure robust variant calling, we applied hard filtering based on several summary statistics. Variants were excluded if they met any of the following criteria: $QD < 2.0$, $QUAL < 30.0$, $SOR > 3.0$, $FS > 60.0$, $MQ < 40.0$, $MQRankSum < -12.5$, $ReadPosRankSum < -8.0$. Further details of these filters and an overview of the workflow are provided in Fig. S1.

The final dataset retained >300,000 SNPs across 276 samples (72 newly sequenced samples from this study and 204 sequences from previously published datasets). To assess the presence of polyclonal infections, we evaluated within-host diversity by calculating $F_{ws}$ coefficients[24]. Empirically, isolates with $F_{ws} < 0.95$ are generally considered as polyclonal, while those with $F_{ws} > 0.95$ are expected to be monoclonal. This analysis revealed that approximately one-third of the isolates (32.4%) could be classified as polyclonal (Tabel S2 and Fig. S2). This finding corroborates previous studies showing that *P. vivax* infections in Southeast Asia and South America are frequently characterized by multiple clones[8,25–28]. For subsequent analyses, polyclonal infections were excluded to avoid biases in heterozygosity, while complementary analyses including both monoclonal and polyclonal infections were conducted and reported in the Supplementary Information to assess the robustness of the results (Fig. S3).

### Global genetic diversity

The population structure of *P. vivax* isolates was characterized by low levels of admixture overall, with clear geographic genetic clusters identified (Fig. 1A). Isolates from South America (Brazil and Colombia) exhibited more ambiguous clustering profiles, suggesting some degree of admixture that requires further investigation with a larger sample size.

The Principal Coordinate Analysis (PCO) corroborated the genetic clustering and revealed four distinct geographic groups (Fig. 1B): an East/West African cluster (Ethiopia, Djibouti, Mauritania), an Indian Ocean cluster (Madagascar, Comoros), an Asian/Pacific Coral Triangle cluster (Cambodia, Indonesia, Malaysia, Papua New Guinea, Thailand, and Vietnam), and a South American cluster (Brazil, Colombia), which appeared centrally positioned relative to the other three clusters.

Among the sub-Saharan African samples, we observed a notable distinction between the two African subpopulations: the East/West African cluster (Ethiopia, Djibouti, Mauritania) and the Indian Ocean cluster (Madagascar, Comoros). Notably, the Indian Ocean cluster was genetically closer to the East/West African cluster than to the Asian/Pacific cluster, suggesting significant historical or contemporary gene flow between these African populations.

This observation contrasts with the assumption of a unique Indonesian origin for the *P. vivax* population in Madagascar[29]. While past human migrations have been proposed as a major driver of *P. vivax* population structure[6,30], our results suggest that the genetic landscape in Madagascar may reflect a more complex history, potentially involving multiple introductions and subsequent gene flow between African and Indian Ocean populations. These findings align partially with previous studies but highlight the need for larger sample sizes and deeper analyses to refine our understanding of the genetic origins and population dynamics of *P. vivax* in Madagascar and the surrounding regions.

### Genomic islands of genetic differentiation

We investigated whether the low frequency of Duffy antigens in human populations across sub-Saharan Africa might exert selective pressure on *P. vivax* parasites, potentially favoring adaptations enabling Duffy-independent invasion pathways. To test this hypothesis, we performed a genome-wide analysis of genetic differentiation using pairwise comparisons of *P. vivax* sequences from multiple regions. We identified eight genomic regions with clear peaks of differentiation on chromosomes 4, 5, 7, 10, 11, 12, 13, and 14 (Fig. 1C). These regions contained genes with diverse functions, including loci involved in metabolic pathways, cellular components, and genes of unknown function (Table 2). Interestingly, we observed signals in three genes previously associated with antimalarial drug resistance: *mdr-1* (PVP01_1010900), *dhfr* (PVP01_0526600), and *dhps* (PVP01_1429500). These findings likely reflect differential drug pressures acting on *P. vivax* populations worldwide, consistent with observations from other studies[25–27]. Importantly, no significant signals were detected in genes encoding parasite ligands directly implicated in erythrocyte invasion pathways, such as PvDBP or members of the PvRBP family, which would have suggested adaptations for Duffy-independent invasion. However, one notable exception was a signal detected in the *thrombospondin-related anonymous protein* gene (TRAP) (PVP01_1218700), located on chromosome 12. This protein is known to be expressed in *P. vivax* sporozoites and plays a critical role in the invasion of human hepatocytes[31,32]. While this signal may not directly relate to erythrocyte invasion, it highlights the potential involvement of *P. vivax* genes in other stages of the parasite's lifecycle. Taken together, our results suggest that the global genomic structure of *P. vivax* populations reflects consistent patterns of differentiation, likely driven by geographic and environmental pressures, including drug pressure. Although no evidence was found for specific genetic adaptations to Duffy-independent invasion in the regions examined, these findings underscore the complexity of *P. vivax* evolution in regions where Duffy negativity predominates. Future studies

**Table 1 | List of *P. vivax* field isolates sequenced in this study**

| Country | Sample | Duffy genotype of patient | Sequence |
|---|---|---|---|
| Angola | ANG71 | Duffy-negative (homozygous) | Not interpretable |
| Angola | ANG83 | Duffy-negative (homozygous) | Not interpretable |
| Angola | ANG168 | Duffy-negative (homozygous) | Not interpretable |
| Angola | ANG172 | Duffy-negative (homozygous) | Not interpretable |
| Angola | ANG189 | Duffy-negative (homozygous) | Not interpretable |
| Angola | ANG192 | Duffy-negative (homozygous) | Not interpretable |
| Angola | ANG193 | Duffy-negative (homozygous) | Not interpretable |
| Angola | ANG201 | Duffy-negative (homozygous) | Not interpretable |
| Angola | ANG207 | Duffy-negative (homozygous) | Not interpretable |
| Angola | ANG209 | Duffy-negative (homozygous) | Not interpretable |
| Angola | ANG210 | Duffy-negative (homozygous) | Not interpretable |
| Angola | ANG212 | Duffy-negative (homozygous) | Not interpretable |
| Burundi | 1803016292 | Duffy-positive (homozygous) | Not interpretable |
| Comoros | 711035003 | Duffy-positive (homozygous) | Interpretable |
| Comoros | 10157 | Duffy-positive (homozygous) | Interpretable |
| Comoros | 11315 | Duffy-positive (heterozygous) | Interpretable |
| Comoros | 11729 | Duffy-positive (heterozygous) | Interpretable |
| Comoros | 8330 | Duffy-positive (heterozygous) | Interpretable |
| Comoros | 10555 | Duffy-positive (heterozygous) | Not interpretable |
| Comoros | 11812 | Duffy-positive (heterozygous) | Not interpretable |
| Comoros | 11994 | Duffy-positive (heterozygous) | Not interpretable |
| Djibouti | 1609055605 | Duffy-positive (homozygous) | Not interpretable |
| Djibouti | 1812066011 | Duffy-positive (homozygous) | Not interpretable |
| Djibouti | 1807003070 | Duffy-positive (homozygous) | Interpretable |
| Egypt | 1408007924 | Duffy-positive (heterozygous) | Not interpretable |
| Ethiopia | 711390488 | Duffy-positive (homozygous) | Interpretable |
| Ethiopia | A1001 | Duffy-negative (homozygous) | Not interpretable |
| Ethiopia | A1002 | Duffy-positive (heterozygous) | Not interpretable |
| Ethiopia | A1004 | Duffy-positive (heterozygous) | Interpretable |
| Ethiopia | A1005 | Duffy-positive (heterozygous) | Interpretable |
| Ethiopia | A1006 | Duffy-positive (heterozygous) | Not interpretable |
| Ethiopia | Ad8001 | Duffy-positive (heterozygous) | Interpretable |
| Ethiopia | Am008 | Duffy-positive (homozygous) | Not interpretable |
| Ethiopia | Am010 | Duffy-positive (heterozygous) | Interpretable |
| Ethiopia | Am012 | Duffy-positive (heterozygous) | Not interpretable |
| Ethiopia | Aw004 | Duffy-positive (heterozygous) | Not interpretable |
| Ethiopia | Aw0005 | Duffy-positive (heterozygous) | Interpretable |
| Ethiopia | Aw0007 | Duffy-positive (heterozygous) | Interpretable |
| Ethiopia | G6001 | Duffy-positive (heterozygous) | Not interpretable |
| Ethiopia | G6003 | Duffy-positive (heterozygous) | Interpretable |
| Ethiopia | G6005 | Duffy-positive (heterozygous) | Interpretable |
| Ethiopia | G6006 | Duffy-positive (heterozygous) | Not interpretable |
| Ethiopia | G6007 | Duffy-positive (heterozygous) | Not interpretable |
| Ethiopia | H9002 | Duffy-positive (heterozygous) | Interpretable |
| Ethiopia | H9003 | Duffy-positive (heterozygous) | Not interpretable |
| Ethiopia | K101 | Duffy-positive (heterozygous) | Interpretable |
| Ethiopia | K102 | Duffy-positive (heterozygous) | Not interpretable |
| Ethiopia | K107 | Duffy-negative (homozygous) | Not interpretable |
| Ethiopia | MC4005 | Duffy-positive (heterozygous) | Interpretable |
| Ethiopia | MC4008 | Duffy-positive (heterozygous) | Not interpretable |
| Ethiopia | MC4010 | Duffy-positive (heterozygous) | Not interpretable |

**Table 1 (continued) | List of _P. vivax_ field isolates sequenced in this study**

| Country | Sample | Duffy genotype of patient | Sequence |
|---|---|---|---|
| Ethiopia | MC4011 | Duffy-positive (heterozygous) | Interpretable |
| Ethiopia | MC4012 | Duffy-positive (heterozygous) | Interpretable |
| Ethiopia | MC4013 | Duffy-positive (heterozygous) | Not interpretable |
| Ethiopia | MC4014 | Duffy-positive (heterozygous) | Not interpretable |
| Ethiopia | MC4016 | Duffy-positive (heterozygous) | Interpretable |
| Ethiopia | MC4018 | Duffy-positive (heterozygous) | Not interpretable |
| Ethiopia | MC4019 | Duffy-positive (heterozygous) | Interpretable |
| Ethiopia | MC4020 | Duffy-positive (homozygous) | Interpretable |
| Ethiopia | MC4021 | Duffy-positive (heterozygous) | Interpretable |
| Ethiopia | MC4022 | Duffy-positive (heterozygous) | Interpretable |
| Ethiopia | MC4023 | Duffy-positive (homozygous) | Interpretable |
| Ethiopia | MC4024 | Duffy-positive (heterozygous) | Interpretable |
| Ethiopia | MC4027 | Duffy-positive (heterozygous) | Interpretable |
| Ethiopia | MC4030 | Duffy-positive (heterozygous) | Not interpretable |
| Ethiopia | MC4031 | Duffy-positive (homozygous) | Interpretable |
| Ethiopia | MC4032 | Duffy-positive (homozygous) | Interpretable |
| Ethiopia | MC4033 | Duffy-positive (heterozygous) | Interpretable |
| Ethiopia | MC4034 | Duffy-positive (heterozygous) | Not interpretable |
| Ethiopia | MC4035 | Duffy-positive (homozygous) | Not interpretable |
| Ethiopia | MC4036 | Duffy-positive (homozygous) | Not interpretable |
| Ethiopia | MC4037 | Duffy-positive (heterozygous) | Interpretable |
| Ethiopia | MC4038 | Duffy-positive (heterozygous) | Interpretable |
| Ethiopia | MC4040 | Not Identified | Interpretable |
| Ethiopia | MC4043 | Duffy-positive (homozygous) | Interpretable |
| Ethiopia | MC4044 | Duffy-positive (heterozygous) | Not interpretable |
| Ethiopia | MC4046 | Duffy-positive (homozygous) | Not interpretable |
| Ethiopia | MC4047 | Duffy-positive (homozygous) | Interpretable |
| Ethiopia | MC4048 | Duffy-positive (heterozygous) | Not interpretable |
| Ethiopia | MC4049 | Duffy-positive (homozygous) | Interpretable |
| Ethiopia | MC4050 | Duffy-positive (heterozygous) | Not interpretable |
| Ethiopia | MC4052 | Duffy-positive (heterozygous) | Interpretable |
| Ethiopia | MC4053 | Duffy-positive (homozygous) | Not interpretable |
| Ethiopia | MJ0016 | Duffy-positive (heterozygous) | Interpretable |
| Ethiopia | MJ0017 | Duffy-positive (heterozygous) | Interpretable |
| Ethiopia | MJ0018 | Duffy-positive (heterozygous) | Interpretable |
| Ethiopia | MJ7001 | Duffy-positive (heterozygous) | Interpretable |
| Ethiopia | MJ7003 | Duffy-positive (heterozygous) | Interpretable |
| Ethiopia | MJ7004 | Duffy-positive (heterozygous) | Interpretable |
| Ethiopia | MJ7006 | Duffy-positive (heterozygous) | Not interpretable |
| Ethiopia | MT0023 | Duffy-negative (homozygous) | Not interpretable |
| Ethiopia | W5011 | Duffy-negative (homozygous) | Not interpretable |
| Eritrea | 11302 | Duffy-positive (heterozygous) | Not interpretable |
| Eritrea | 1801078268 | Duffy-positive (heterozygous) | Not interpretable |
| Mauritania | 711M21044701 | Duffy-positive (homozygous) | Interpretable |
| Mauritania | 713285008 | Duffy-positive (homozygous) | Interpretable |
| Mauritania | 714035002 | Duffy-positive (homozygous) | Interpretable |
| Mauritania | 714145012 | Duffy-positive (heterozygous) | Not interpretable |
| Mauritania | 11034 | Duffy-positive (heterozygous) | Not interpretable |
| Mauritania | MAU24 | Duffy-negative (homozygous) | Not interpretable |
| Mauritania | MAUAT68 | Duffy-negative (homozygous) | Not interpretable |
| Madagascar | MAE-CSB01-001 | Duffy-positive (homozygous) | Interpretable |

**Table 1 (continued) | List of *P. vivax* field isolates sequenced in this study**

| Country | Sample | Duffy genotype of patient | Sequence |
|---|---|---|---|
| Madagascar | MAE-CSB01-002 | Duffy-positive (heterozygous) | Interpretable |
| Madagascar | MAE-CSB02-001 | Duffy-positive (heterozygous) | Interpretable |
| Madagascar | MAE-CSB02-002 | Duffy-positive (heterozygous) | Interpretable |
| Madagascar | MAE-CSB02-003 | Duffy-positive (heterozygous) | Interpretable |
| Madagascar | MAE-CSB03-002 | Duffy-positive (heterozygous) | Interpretable |
| Madagascar | MAE-CSB03-003 | Duffy-positive (heterozygous) | Interpretable |
| Madagascar | MAE-V01-001 | Duffy-positive (heterozygous) | Interpretable |
| Madagascar | MAE-V01-002 | Not identified | Interpretable |
| Madagascar | MDZ-V02-001 | Duffy-positive (homozygous) | Not interpretable |
| Madagascar | MDZ-V02-006 | Duffy-positive (homozygous) | Not interpretable |
| Madagascar | MDZ-V03-001 | Duffy-positive (homozygous) | Not interpretable |
| Madagascar | MAE-V04-001 | Duffy-positive (heterozygous) | Interpretable |
| Madagascar | MAE-V06-001 | Duffy-positive (heterozygous) | Interpretable |
| Madagascar | MDZ-CSB01-001 | Duffy-positive (heterozygous) | Interpretable |
| Madagascar | MDZ-CSB01-002 | Duffy-positive (heterozygous) | Interpretable |
| Madagascar | MDZ-CSB01-003 | Duffy-positive (heterozygous) | Interpretable |
| Madagascar | MDZ-CSB01-005 | Duffy-positive (heterozygous) | Not interpretable |
| Madagascar | MDZ-V01-001 | Duffy-positive (heterozygous) | Interpretable |
| Madagascar | MDZ-V02-001 | Duffy-positive (homozygous) | Interpretable |
| Madagascar | MDZ-V02-002 | Duffy-positive (heterozygous) | Interpretable |
| Madagascar | MDZ-V02-004 | Duffy-positive (heterozygous) | Interpretable |
| Madagascar | MDZ-V02-005 | Duffy-positive (heterozygous) | Interpretable |
| Madagascar | MDZ-V02-006 | Duffy-positive (homozygous) | Interpretable |
| Madagascar | MDZ-V04-001 | Duffy-positive (heterozygous) | Interpretable |
| Madagascar | MDZ-V04-002 | Duffy-positive (heterozygous) | Interpretable |
| Madagascar | MDZ-V04-003 | Duffy-positive (heterozygous) | Interpretable |
| Madagascar | 711185003 | Duffy-positive (heterozygous) | Interpretable |
| Madagascar | 713455019 | Duffy-positive (homozygous) | Interpretable |
| Sudan | 713405019 | Duffy-positive (homozygous) | Not interpretable |
| Sudan | 713485020 | Duffy-positive (homozygous) | Not interpretable |
| Sudan | 510m15000201 | Duffy-positive (heterozygous) | Not interpretable |

incorporating larger datasets of high-quality genomes from Duffy-negative individuals will be essential to conclusively identify loci under selection for alternative invasion pathways.

**Genetic diversity of genes associated with drug resistance**
In the African *P. vivax* population, we detected four single non-synonymous mutations (P33L, C49R, N130K, A255T) in the *bifunctional dihydrofolate reductase-thymidylate synthase* gene (PVP01_0526600) in samples from Madagascar, Comoros, and Mauritania. The C49R mutation, only found in Madagascar (18/23), was the most common mutation (26%). The P33L mutation was detected in samples from the Comoros (1/6), the N130K mutation in samples from Madagascar (3/23) and the A255T mutation in samples from Mauritania (1/3). No such mutations were found in samples from Ethiopia and Djibouti.

Analysis of mutation points in the *hydroxymethyldihydropterin pyrophosphokinase-dihydropteroate synthase* gene (PVP01_1429500), likely associated with sulfadoxine resistance, revealed the presence of five different non-synonymous mutations (E142G, M205I, G383A, I545T and A647V) in varying proportions. The three most common mutations were G383A (36/68) from 4 countries, followed by M205I (30/68) from 3 countries, and E142G (25/68) from 2 countries. The E142G, I545T and A647V mutations were country-specific: E142G in Ethiopia and Djibouti (24/34 and 1/1, respectively), I545T in Mauritania (1/3) and A647V in the Comoros (1/6)

and Ethiopia (10/34). The M205I and G383A mutations were found in varying proportions in all countries except in samples from Madagascar in which we did not find M205I mutations. Three triple mutant alleles were observed including the E142G/M205I/G383A (15/68, mainly in Ethiopia 14/15), the E142G/M205I/A647V (10/68, only in Ethiopia 10/34) and the M205I/G383A/I545T (1/68, Mauritania). We also found two double mutant alleles, the M205I/G383A (4/68) and the G383A/A647V (1/68).

For the *ABC transporter B family member 1 - multidrug resistance protein 1* gene (PVP01_1010900), which is thought to modulate parasite susceptibility to amino-4 and amino-alcohol quinolines, five-point mutations were found (F194Y, S698G, L845F, F976Y and T1269S). The F976Y mutation was the most common mutation (34/68) and was found in high proportions in Mauritania (3/3), in Ethiopia (30/34) and in Djibouti (1/1). The F194Y mutation was found only in samples from Madagascar (1/24), the S698G mutation was observed in samples from Mauritania (1/3) and Ethiopia (1/34) and the L845F mutation only in Mauritania (2/3). Two double mutant alleles were observed including the S698G/F976Y (7/68, mainly in Ethiopia 6/34) and the L845F/F976Y (2/68, Mauritania 2/3). More details are given in the Supplementary Information (Table S3).

**Genetic diversity of invasion-related genes**
We then restricted our analysis to regions containing genes validated or suspected to encode parasite ligands involved in reticulocyte invasion

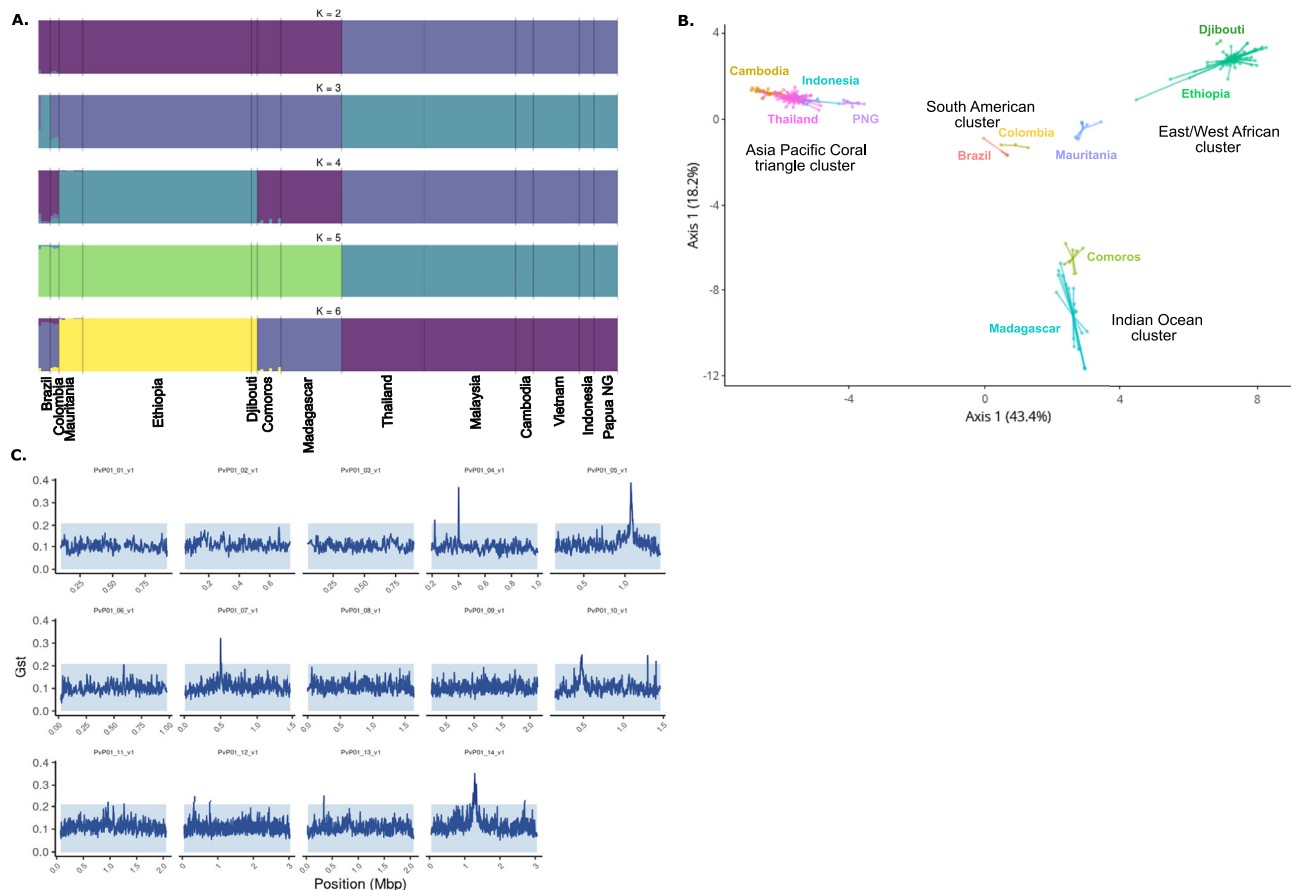

**Fig. 1 | Global genetic diversity of *P. vivax* populations. A** Structural analysis of chromosomes. **B** Principal Coordinate analysis (PCO). **C** Genome-wide genetic differentiation with genomic islands.

pathways[25–27,33], such as the *pvdbp*, *pvebp/pvdbp2*, *pvrbp2a*, and *pvrbp2b* genes (Table S4). Only two mutations were found in the *P. vivax Duffy binding* gene (*pvdbp*, PVP01_0623800), the K277T (2/68) and the G830V (1/68). The K277T mutation was specific to samples from the Comoros (1/6) and Madagascar (1/24), while the G830V mutation was detected only in Mauritania (1/3). No mutations were detected in Ethiopia and Djibouti. Analysis of mutation points in the *P. vivax erythrocyte binding* gene (*pvebp/pvdbp2*, PVP01_0102300) revealed 7 nonsynonymous mutations: D268N, E341K, K595N, I611F, E660K, V705L and F746I. The most common mutation was the I611F (8/68), found only in Comoros (1/6) and Madagascar (7/24). In Ethiopia, only the E660K mutation was detected at a low proportion (1/34), while no mutation was found in samples from Djibouti. The K595N mutation was specific for Madagascar (1/24) and the V705L mutation for the Comoros (1/6). The D268N mutation was found in Comoros (1/6), in Madagascar (1/24) and in Mauritania (1/3). Two double mutant alleles were found in Comoros and Madagascar (D268N/E341K) and in Mauritania (D268N/F746I).

No mutations were found in the *reticulocyte binding protein 2a* (*pvrbp2a*, PVP01_1402400) and the *reticulocyte binding protein 2b* (*pvrbp2b*, PVP01_0800700) genes in samples from Mauritania, Ethiopia, and Djibouti. In samples from the Comoros, the K112I mutation in the *pvrbp2a* gene and the S186N and D461G mutations in the *pvrbp2b* gene were detected once (1/6). The same mutations were found in samples from Madagascar, once for the K112I mutation in the *pvrbp2a* gene (1/24) and at higher frequencies for the S186N (6/24) and D461G (3/24) mutations in the *pvrbp2b* gene. The L84F in the *pvrbp2b* gene was unique to samples from Madagascar (5/24). More details can be found in the Supplementary Information (Table S4).

We also analyzed genetic differentiation by estimating Tajima's *D* values at loci associated with invasion-related genes and comparing these values with those obtained for all individual genes across the genome. This analysis revealed that invasion-related genes from *P. vivax* populations in all regions fell well within the boundaries defined by other genes (Fig. 2A). In addition, a country-level comparison of Tajima's *D* values across 13 populations confirmed that invasion-related genes did not exhibit patterns of differentiation distinct from the genome-wide average. Interestingly, values of Tajima's D across all populations appeared shifted toward negative values, suggesting an excess of low-frequency alleles. This trend may indicate ongoing geographic differentiation at a finer scale, possibly driven by local demographic events, such as population expansions or recent bottlenecks, rather than by selective pressure on invasion-related loci.

Overall, we found no evidence of selective pressure specifically acting on invasion-related genes in sub-Saharan *P. vivax* strains (Fig. 2B, C). These results suggest that while geographic structure may influence allele frequency patterns, it does not appear to impose measurable selection on genes linked to erythrocyte invasion pathways.

### Genetic diversity associated with Duffy genotype in human hosts
We hypothesized that the Duffy status of patients (homozygous and heterozygous Duffy-positives, and homozygous Duffy-negative) might influence the diversity and selection of invasion-related genes in African *P. vivax* genomes. To test this, we conducted a Genome-Wide Association Study (GWAS) on sequences obtained from 18 isolates infecting homozygous Duffy-positive patients and 52 isolates from heterozygous Duffy-positive patients. The results of the GWAS showed no significant associations between the patient's Duffy status and parasite genotypes at loci encoding invasion-related genes. Specifically, we did not observe any molecular

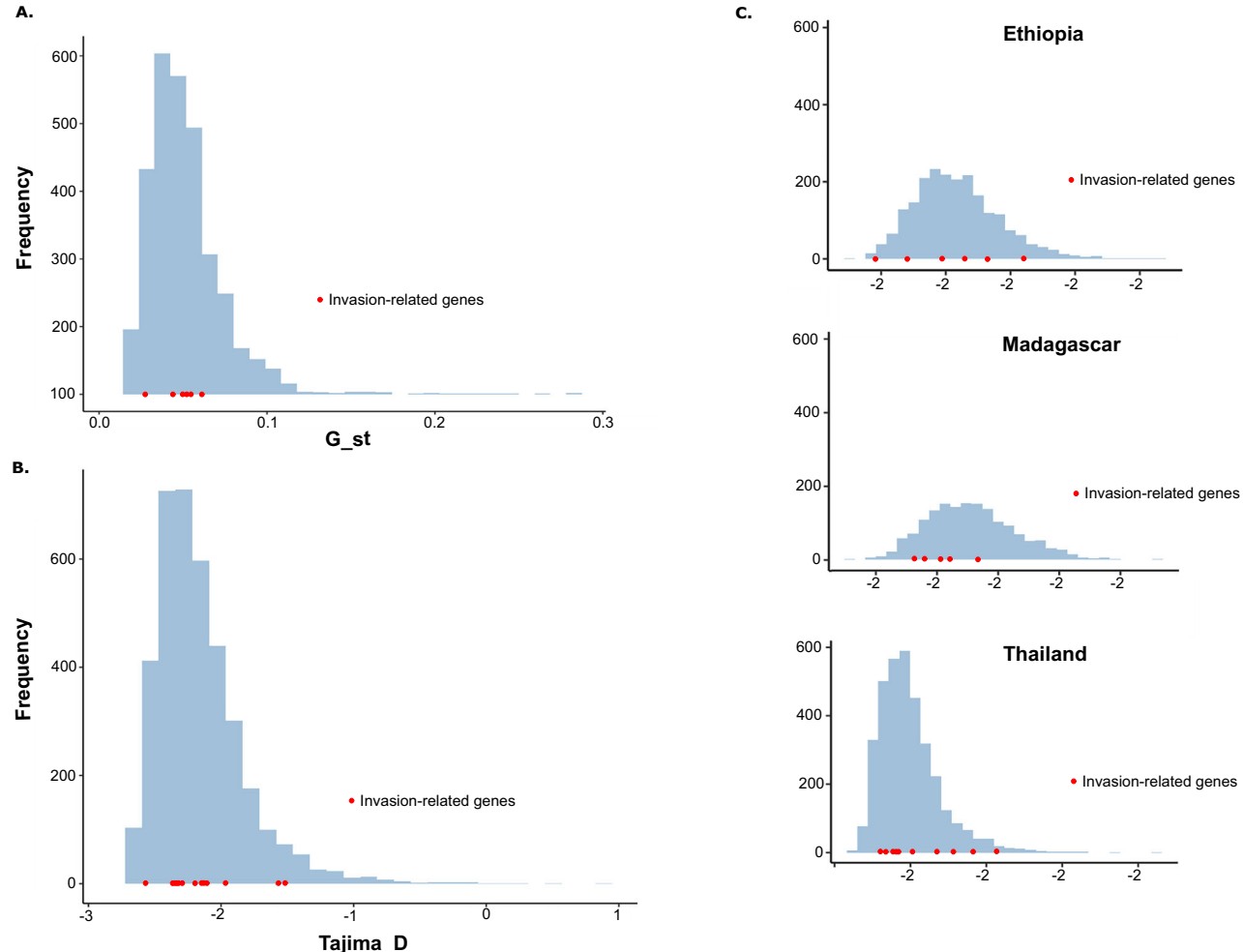

**Fig. 2 | Genetic diversity of invasion-related genes. A** Differentiation of invasion-related genes in all *P. vivax* populations. **B** Analysis of Tajima's D within all countries. **C** Analysis of Tajima's D in Ethiopia, Madagascar, and Thailand.

signatures suggestive of adaptive evolution in these genes that could facilitate infection in Duffy-negative hosts (Fig. 3). These findings suggest that the evolution of African *P. vivax* populations may not be primarily driven by diversification or selection of invasion-related genes, at least within the resolution of our current analysis. Alternatively, adaptations to infect Duffy-negative individuals, if they exist, may involve molecular mechanisms too complex or subtle to be detected by broad-scale genomic association methods, particularly given the constraints imposed by the limited availability of high-quality Duffy-negative sequences.

Overall, while our data do not rule out the possibility of Duffy-independent invasion pathways, they underscore the need for future studies incorporating higher-resolution genomic data from larger numbers of Duffy-negative infections to identify potential loci under selection and fully elucidate the evolutionary dynamics of *P. vivax* populations in Africa.

## Discussion

To date, the global genetic diversity and population structure of *P. vivax* remain poorly understood, particularly in Africa, where vivax malaria is rare and characterized by low parasitemia. While previous studies have demonstrated distinct population structures in Southeast Asia and South America, limited genomic data exist for *P. vivax* populations circulating in Africa, except for southern Ethiopia. Here, using genomic data from African and global isolates, we provide a comparative analysis of *P. vivax* populations and explore the potential impact of Duffy negativity on parasite evolution.

Our results confirm the clear geographic genetic structure of *P. vivax*, distinguishing African, Asian, and South American clusters, consistent with prior studies. Within Africa, we observed two distinct sub-clusters: the East/West African cluster (Ethiopia, Djibouti, Mauritania) and the Indian Ocean cluster (Madagascar and Comoros). Notably, the Indian Ocean cluster was genetically closer to East Africa than to Asia, challenging previous assumptions of a strictly Indonesian origin for the Malagasy *P. vivax* population. While earlier work suggested single origins, our findings, combined with the complex demographic history of Madagascar, indicate multiple introductions likely shaped by human migrations[6,30].

Our findings align with previous observations reported by Hupalo et al.[26], which demonstrated a clear geographic genetic structure of *P. vivax* populations worldwide, including low levels of admixture. Similarly, our analysis revealed well-defined clusters for African, Asian, and South American populations, with limited gene flow between regions. In addition, the work of Benavente et al.[25] identified loci under selective pressure, particularly in drug resistance genes (*pvkelch10*, *pvmrp1*), which differ from the loci identified in our study (*pvmdr1*, *pvdhfr*, *pvdhps*). This discrepancy likely reflects regional variations in drug use and the selective pressures acting on *P. vivax* populations in Africa compared to other regions.

A key hypothesis motivating this study was that Duffy-negativity might impose selective pressure on invasion-related genes, driving adaptations in *P. vivax* African populations. Genetic diversity within major invasion-related loci did not deviate significantly from the genome-wide background, and Tajima's D analyses confirmed a lack of positive selection. For example,

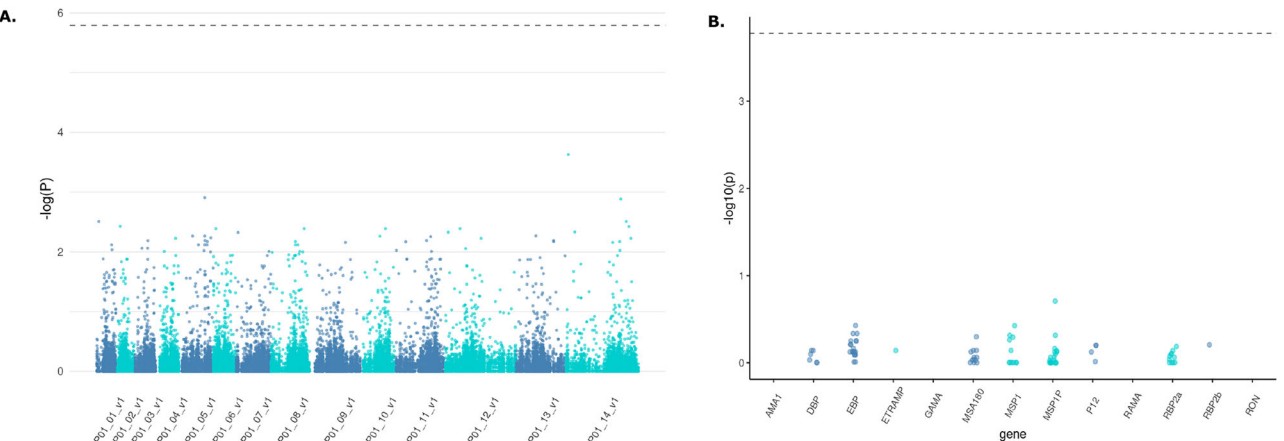

**Fig. 3 | GWAS analysis. A** GWAS analysis of sequences obtained from 18 *P. vivax* African isolates infecting homozygous Duffy-positive patients and 52 *P. vivax* African isolates infecting heterozygous Duffy-positive patients for invasion-related genes. Manhattan plot displaying the genome-wide association results. Each dot represents a single nucleotide polymorphism (SNP), plotted as the negative log10-transformed *p* value $(-\log10(P))(-\log_{10}(P))(-\log(P))$ of the association test. The x-axis represents genomic regions, labeled as PvP01_01_v1 through PvP01_14_v1, corresponding to different chromosomes or contigs of the

*Plasmodium vivax* genome. The dashed horizontal line indicates the genome-wide significance threshold. No SNPs surpassed the genome-wide significance threshold, suggesting that no variants showed statistically significant associations after multiple testing correction. **B** Association of genetic variants in selected genes with the phenotype of interest. Genes are displayed on the x-axis, and the dashed horizontal line indicates the genome-wide significance threshold. Genes include *AMA1*, *DBP*, *EBP*, *ETRAMP*, *GAMA*, *MSA180*, *MSP1*, *MSP1P*, *P12*, *RAMA*, *RBP2a*, *RBP2b*, and *RON*. No SNPs reached the genome-wide significance threshold in this analysis.

the *pvdbp* gene showed only two novel non-synonymous mutations (K277T and G830V), while mutations in *pvebp/pvdbp2* and *pvrbp2b* were observed at low frequencies without clear evidence of selection.

As a result, potential adaptive signals may remain undetected, particularly if they involve subtle or complex evolutionary mechanisms.

Recent studies suggest that Duffy-negative erythroblasts transiently express DARC during terminal differentiation, providing a limited opportunity for *P. vivax* merozoites to invade[15,34]. This biological constraint could explain the low parasitemia consistently observed in Duffy-negative individuals and the apparent lack of strong selection signals in invasion-related genes. If alternative invasion pathways exist, they may rely on molecular mechanisms that are currently undetectable with our analytical approach or require larger, more balanced datasets for validation.

While no evidence of selection was observed in invasion-related genes, we identified strong signals of selection in genes associated with antimalarial drug resistance, including *pvdhfr*, *pvdhps*, and *pvmdr1*. These results are consistent with earlier studies, such as Benavente et al.[25], which reported signals of antifolate resistance markers. However, our findings differ in specific loci under selection, as we detected positive selection in *pvmdr1* (not observed by Benavente et al.), while signals in *pvkelch10* and *pvmrp1* were absent in our dataset[25]. These discrepancies likely reflect region-specific drug pressures and variations in antimalarial use across African and global populations. Notably, the detection of previously unreported mutations, such as I545T in *pvdhps* and F194Y in *pvmdr1*, highlights the ongoing evolution of drug resistance in African *P. vivax* populations.

Despite these advances, analysis of the *P. vivax* genome remains technically challenging due to field samples with low parasitemia and high human DNA contamination. While sWGA prior to WGS improved parasite genome recovery, it was insufficient to obtain high-quality sequences from Duffy-negative patients. Moreover, the sWGA approach precluded the detection of copy number variations in key invasion-related genes, such as *pvdbp* and *pvebp/pvdbp2*, which may play critical roles in alternative invasion pathways[35].

Despite increasing evidence of *P. vivax* infections in Duffy-negative individuals, our study was unable to identify clear genomic signatures of parasite adaptation. This limitation primarily stems from the low parasitemia observed in these individuals, which restricted the availability of high-quality genomic sequences. As a result, potential adaptive mechanisms may remain undetected, particularly if they involve complex

or subtle evolutionary processes. Overcoming these challenges will require larger, well-balanced datasets, improved parasite enrichment techniques, and functional studies to elucidate alternative invasion pathways.

In summary, this study provides new insights into the genetic diversity and structure of African *P. vivax* populations and places them within a global context. We confirm the distinct clustering of African *P. vivax* populations and highlight their divergence from Asian and South American populations. However, we found no evidence of selective pressure on invasion-related genes, suggesting that adaptations to Duffy-negative hosts, if they exist, may involve mechanisms too complex to be detected by our analysis. These results emphasize the biological and technical challenges of studying *P. vivax* in Duffy-negative individuals.

## Methods
### Biological samples
All biological samples used in this study were obtained from *Plasmodium vivax*-infected blood samples collected between 2016 and 2021 after obtaining informed consent from all participants. The study protocol and sample collection were approved by the Institutional Review Board of Addis Ababa University (Ethiopia), the Comité d'Éthique Biomédicale de Madagascar (Madagascar), the Comité National d'Éthique pour la Recherche en Santé de Mauritanie (Mauritania), and the Comité de Protection des Personnes Île-de-France (France) for samples from returning travelers. All ethical regulations relevant to human research participants were followed. Ethics approval was obtained from an ethics committee located in Angola for the sample collection that took place in Angola. Dried blood spots and/or veinous blood samples were collected from symptomatic patients at health centers in Ethiopia (Addis Ababa University), Madagascar (Institut Pasteur Madagascar), and Mauritania (Université de Nouakchott) and from symptomatic travelers returning from Comoros, Djibouti, Madagascar, Mauritania, and Ethiopia (French National Reference Center for malaria). Blood samples from Madagascar were leukodepleted using CF11-packed columns to minimize the amount of human DNA[36]. We enriched our dataset with 204 published genomic sequences of *P. vivax* strains circulating in Southeast Asia (Cambodia, Thailand, Vietnam), Pacific Coral Triangle (Indonesia, Malaysia, Papua New Guinea), South America (Brazil, Columbia), and Africa (Ethiopia, Madagascar, Mauritania) (Table 1 and Table S1).

**Table 2 | List of genes falling within the identified genomic islands showing clear peaks of differentiation in eight genomic regions on chromosomes 4, 5, 7, 10, 11, 12, 13, and 14**

| Chr. | Pos. start | Pos. end | ID | Description | Function | |
|---|---|---|---|---|---|---|
| 4 | 219558 | 220885 | PVP01_0404900 | Plasmodium exported protein, unknown function | Unknown | |
| 4 | 399586 | 401784 | PVP01_0409900 | acyl-CoA synthetase, putative | Metabolic pathways | Fatty acid biosynthesis and degradation |
| 5 | 1061019 | 1067947 | PVP01_0526300 | conserved Plasmodium protein, unknown function | Unknown | |
| 5 | 1070413 | 1071498 | PVP01_0526400 | conserved Plasmodium protein, unknown function | Unknown | |
| 5 | 1072310 | 1074121 | PVP01_0526500 | mRNA-binding protein PUF2, putative | Metabolic pathways | RNA binding |
| 5 | 1077362 | 1079236 | PVP01_0526600 | bifunctional dihydrofolate reductase-thymidylate synthase, putative | Metabolic pathways | Folate biosynthesis/Drug resistance |
| 5 | 1080118 | 1082726 | PVP01_0526700 | LETM1-like protein, putative | Metabolic pathways | ribosome binding |
| 5 | 1085714 | 1102054 | PVP01_0526800 | conserved Plasmodium protein, unknown function | Unknown | |
| 7 | 495347 | 505297 | PVP01_0709800 | cysteine repeat modular protein 1, putative | Cellular component | integral component of membrane |
| 10 | 470947 | 474113 | PVP01_1010700 | heptatricopeptide repeat-containing protein, putative | Unknown | |
| 10 | 475543 | 477329 | PVP01_1010800 | cytochrome b-c1 complex subunit 2, putative | Metabolic pathways | protein processing involved in protein targeting to mitochondrion |
| 10 | 478739 | 483133 | PVP01_1010900 | ABC transporter B family member 1, putative | Metabolic pathways | transmembrane transport, food vacuole |
| 10 | 1302679 | 1303648 | PVP01_1030200 | 60S ribosomal protein L31, putative | Metabolic pathways | translation |
| 11 | 961821 | 964757 | PVP01_1121700 | acetyl-CoA synthetase, putative | Metabolic pathways | acetyl-CoA biosynthetic process |
| 12 | 323534 | 324835 | PVP01_1208000 | 6-cysteine protein P47 | Cellular component | cytoplasm, cell surface |
| 12 | 768961 | 770631 | PVP01_1218700 | thrombospondin-related anonymous protein, putative | Metabolic pathways | entry into host, protein binding (sporozoite stages) |
| 12 | 771355 | 773349 | PVP01_1218800 | conserved Plasmodium protein, unknown function | Unknown | |
| 13 | 332331 | 342481 | PVP01_1307300 | cysteine repeat modular protein 3, putative | Unknown | |
| 14 | 1225902 | 1230319 | PVP01_1428700 | conserved protein, unknown function | Unknown | |
| 14 | 1231284 | 1233767 | PVP01_1428800 | histone-arginine methyltransferase CARM1, putative | Metabolic pathways | histone methylation |
| 14 | 1235239 | 1237877 | PVP01_1428900 | conserved protein, unknown function | Unknown | |
| 14 | 1242635 | 1248727 | PVP01_1429000 | CCR4-associated factor 1, putative | Metabolic pathways | nucleic acid binding |
| 14 | 1254473 | 1258020 | PVP01_1429100 | ER membrane protein complex subunit 1, putative | Metabolic pathways | protein folding in endoplasmic reticulum |
| 14 | 1259368 | 1261188 | PVP01_1429200 | mitochondrial carrier protein, putative | Unknown | |
| 14 | 1262207 | 1264847 | PVP01_1429300 | cullin-1, putative | Metabolic pathways | ubiquitin-dependent protein catabolic process |
| 14 | 1267481 | 1268779 | PVP01_1429400 | conserved Plasmodium protein, unknown function | Unknown | |
| 14 | 1269756 | 1272304 | PVP01_1429500 | hydroxymethyldihydropterin pyrophosphokinase-dihydropteroate synthase, putative | Metabolic pathways | Folate biosynthesis/Drug resistance |
| 14 | 1273133 | 1274237 | PVP01_1429600 | conserved Plasmodium protein, unknown function | Unknown | |
| 14 | 1276317 | 1279088 | PVP01_1429700 | ATP-dependent RNA helicase DBP1, putative | Metabolic pathways | nucleic acid binding |
| 14 | 1284812 | 1287310 | PVP01_1429800 | protein phosphatase PPM7, putative | Metabolic pathways | protein dephosphorylation |
| 14 | 1287683 | 1289728 | PVP01_1429900 | aquaporin, putative | Metabolic pathways | transmembrane transport |
| 14 | 1294483 | 1297856 | PVP01_1430000 | protein phosphatase PPM5, putative | Metabolic pathways | protein dephosphorylation |
| 14 | 1299004 | 1301880 | PVP01_1430100 | ABC1 family, putative | Metabolic pathways | Purine metabolism |
| 14 | 1302963 | 1303546 | PVP01_1430200 | ribosomal protein L33, apicoplast, putative | Metabolic pathways | translation |

**Table 2 (continued) | List of genes falling within the identified genomic islands showing clear peaks of differentiation in eight genomic regions on chromosomes 4, 5, 7, 10, 11, 12, 13, and 14**

| Chr. | Pos. start | Pos. end | ID | Description | Function | |
|------|-----------|----------|-----|-------------|----------|---|
| 14 | 1310045 | 1316657 | PVP01_1430400 | JmjC domain-containing protein, putative | Metabolic pathways | Biosynthesis of secondary metabolites |
| 14 | 1318305 | 1322540 | PVP01_1430500 | conserved Plasmodium protein, unknown function | Unknown | |
| 14 | 1323658 | 1326144 | PVP01_1430600 | RuvB-like helicase 1, putative | Metabolic pathways | Biosynthesis of various antibiotics and secondary metabolites |
| 14 | 1328328 | 1344272 | PVP01_1430700 | peptidase family C50, putative | Metabolic pathways | proteolysis |
| 14 | 2694842 | 2697905 | PVP01_1462600 | conserved Plasmodium protein, unknown function | Unknown | |

### DNA extraction

Genomic DNA was extracted from 100 μL of red blood cell pellets or from a 6-mm dried blood spot punch using the QIAamp mini-blood DNA kit (Qiagen), according to the supplier's instructions. The total DNA concentration was quantified using a dsDNA HS assay kit and a Qubit fluorometer (Invitrogen).

### Screening of *Plasmodium* species

The *Plasmodium* species screening of *P. vivax* isolates was performed by real-time PCR as previously described[37] (Table S5).

### Duffy genotyping

The Duffy genotype of the patients was determined by PCR and Sanger sequencing as previously described[6] (Table S5).

### Selective whole-genome amplification

sWGA was performed to enrich *P. vivax* DNA as previously described[35] (Table S5). PCR products obtained from sWGA were diluted (1:1) with DNase-free and RNase-free water (ThermoFisher Scientific) and purified with AMPure XP beads (Beckman Coulter) according to the supplier's recommendations.

### Whole genome sequencing

Whole genome sequencing was performed as previously described[5]. Sample libraries were prepared using a Truseq Nano Kit (Illumina). Briefly, 100–150 ng of DNA was subjected to shearing, end repair, A-tailing, and adapter ligation. When necessary, they were enriched using 10–15 cycles of PCR. The sample libraries were then multiplexed and loaded. Paired-end sequencing was performed with 2 × 150 base reads kit on a NovaSeq6000. Illumina sequence reads were submitted to an archived system.

### Variant calling

Sequencing reads were aligned to the *Plasmodium vivax* PvP01 reference genome (version 48, PlasmoDB), which is publicly available at https://plasmodb.org/plasmo/[38].

Low-quality reads (MAPQ < 30), secondary alignments, and anomalous insert sizes (>1000 bp) were excluded. Duplicates were marked using Picard. SNPs were called with GATK HaplotypeCaller (v4.1.7.0) in diploid mode, and joint genotyping was performed using GenotypeGVCFs[39]. Variants were filtered using VariantFiltration with the following parameters: QUAL < 30, QD < 2.0, MQ < 40, MQRankSum <–12.5, SOR > 3.0, FS > 60.0, and ReadPosRankSum <–8.0. Genotypes with depth ≤4 were considered missing. SNPs with >10% missing data or not biallelic were excluded using VCFtools, and final VCFs were merged with Picard GatherVcfs. Additional technical details are provided in the *Supplementary Methods*.

### Analysis of genotypic data

**Infection complexity and filtering.** To distinguish between mono- and polyclonal *P. vivax* infections, we computed the within-host diversity index Fws across loci using the formula $F_{ws} = 1 - H_w/H_s$, where $H_w$ is the observed heterozygosity within samples and $Hs$ is the expected heterozygosity at the population level. Given the bias introduced by mixed infections in measures of diversity, structure, and genotype–phenotype associations, we restricted the main analyses to monoinfections. To evaluate the robustness of this choice, we conducted parallel analyses including all infections, whose results were consistent with those from the monoinfection subset (Fig. S3).

**Population structure and genetic clustering.** Genetic population structure was investigated using principal coordinate analysis (PCoA) based on pairwise genetic distances between samples, implemented in R (v4.3.0) with the *ade4* package. Clustering analysis was performed using *fastSTRUCTURE*, a variational Bayesian algorithm, across a range of $K$ values ($K = 2$–$10$)[40]. The most likely number of clusters was inferred using the *chooseK* function. These analyses were restricted to monoinfections. Further methodological details are provided in the *Supplementary Methods*.

**Genomic differentiation and selection.** Genomic islands of differentiation were identified by computing Nei's coefficient of genetic differentiation (GST) for each SNP across populations. To explore selective signatures, we calculated Tajima's D for all genes and focused on loci involved in reticulocyte invasion pathways. Gene-level estimates of genetic differentiation were also computed using GST. These analyses aimed to identify regions potentially under selection in association with host Duffy phenotype or geography.

**Genotype–phenotype association analysis.** To test for associations between parasite genotypes and host Duffy phenotype, we applied a generalized linear model (GLM) with a binomial distribution: Duffy_status = parasite_genotype + country_of_origin. $P$ values associated with genotype effects were computed and corrected for multiple comparisons using the Bonferroni method. Only monoinfections with available Duffy genotyping were included in this analysis.

### Statistics and reproducibility

This study is based on 133 *P. vivax* field isolates collected from ten sub-Saharan African countries and 204 publicly available genomes from other endemic regions. All analyses were conducted on monoinfections (as defined by Fws > 0.95), unless otherwise specified. Replicates correspond to independent patient-derived parasite isolates. No experimental replication was possible due to the observational and retrospective nature of the dataset. Statistical analyses were performed in R (v4.3.0), using appropriate packages and functions as described in the corresponding sections. Significance thresholds and multiple-testing corrections (Bonferroni) were applied where relevant. Genetic diversity, population structure, differentiation, and genotype–phenotype association were assessed with standard population genetics metrics and GLMs. Consistency of results was assessed by including and excluding

polyclonal infections (Fig. S3). Full details of the analytical pipelines are provided in the Supplementary Methods.

## Data availability

All raw sequence data is available from the European Nucleotide Archive (www.ebi.ac.uk/ena; see Supplementary Data 1 for accession numbers). These data include samples generated as part of this study (PRJEB81739) as well as publicly available sequences. Specifically, the dataset incorporates sequences from the LSHTM returning travelers project (PRJEB44419) and the MalariaGEN *P. vivax* Genome Variation project[23,27]. It is important to clarify that while our study includes sequences from the publicly available MalariaGEN dataset, our samples were independently collected and were not generated as part of MalariaGEN activities[41].

## Code availability

All custom R and SLURM scripts used for the analysis of *Plasmodium falciparum* WGS data (BioProject PRJEB81739) are available on Zenodo: https://doi.org/10.5281/zenodo.15391778. The archive includes all scripts used for variant filtering, diversity metrics (e.g., Tajima's D), population structure, GWAS, and HPC workflows (SLURM submission). All files are versioned and released under the MIT license.

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

## Acknowledgements

We thank all patients who contributed blood samples, their guardians, and all team members in the respective health centers of the participating countries. We acknowledge the help of the HPC Core Facility of the Institut Pasteur for this work as well as the Biomics Platform, C2RT, Institut Pasteur, Paris, France (supported by France Génomique, ANR-10-INBS-09, and IBISA). This study was supported by grants overseen by the Human Heredity and Health in Africa (H3Africa, grant H3A/18/002), the French National Research Agency (VIPeRs, ANR-18-CE15-0026, VICTOR ANR- ANR-21-CE35-0006 and PvINV ANR-21-CE15-0013), the Institut Pasteur, Paris, the Laboratoire d'Excellence (LabEx) 'French Parasitology Alliance for Health Care' (ANR-11-15 LABX-0024-PARAFRAP). IB received financial support from the French National Research Agency (VIPeRs, ANR-18-CE15-0026), the FRM (Fondation pour la Recherche Médicale, FRM FDT202204014872), and the Fondation Thérèse Lebrasseur (DM was the winner of the Fondation Thérèse Lebrasseur award in 2021).

## Author contributions

I.B., P.C., F.L., C.C., and D.M. conceived and directed the project. M.S., I.V.-W., A.R., L.G., A.O.M.S.B., S.H., and M.F.F.C. organized the sample collection and processing. I.B. and L.M. performed the laboratory work, including sequencing. I.B. performed bioinformatic analysis under the supervision of P.C., I.B., P.C., and D.M. together interpreted the results. Additional advice from F.L. and C.C. was sought during the analysis. I.B. wrote the first draft of the manuscript under guidance from P.C. and D.M. All the authors commented on the version of the manuscript and approved the final manuscript. I.B., P.C., and D.M. compiled the final version of the manuscript.

## Competing interests

The authors declare no competing interests.

## Additional information

[1]Institut Pasteur, Université Paris Cité, Malaria Genetic and Resistance Unit, INSERM U1201, Paris, France. [2]École Doctorale ED515 « Complexité du vivant », Sorbonne Université, Paris, France. [3]Institut Pasteur, Université Paris Cité, Malaria Parasite Biology and Vaccines Unit, Paris, France. [4]Aklilu Lemma Institute of Pathobiology, Addis Ababa University, Addis Ababa, Ethiopia. [5]Institut Pasteur de Dakar, Dakar, Senegal. [6]Immunology of Infectious Diseases, Institut Pasteur Madagascar, Antananarivo, Madagascar. [7]Faculté de Médecine, Université de Fianarantsoa, Fianarantsoa, Madagascar. [8]Université de Nouakchott, Unité de recherche génomes et milieux, Nouakchott, Mauritania. [9]Fundacão Oswaldo Cruz, Malaria Research Laboratory, Rio de Janeiro, Brazil. [10]Centre National de Référence du Paludisme, Hôpital Bichat, Paris, France. [11]Institut Pasteur, Université Paris Cite, Biomics Platform, Université Paris Cite, Biomics Platform, C2RT, Paris, France. [12]Department of Pathogenic Biology and Immunology, School of Medicine, Key laboratory of Jiangsu province university for Nucleic Acid & Cell Fate Manipulation, Affiliated Hospital of Yangzhou University, Yangzhou University, Yangzhou, China. [13]Institut Pasteur, Hub de Bio-informatiques et Biostatistique, Département Biologie Computationnelle, Université Paris Cité, USR 3756, Paris, France. [14]Université de Strasbourg, UR3073—PHAVI—Pathogens Host Arthropods Vectors Interactions Unit, Strasbourg, France. [15]CHU Strasbourg, Laboratory of Parasitology and Medical Mycology, Strasbourg, France. [16]Institut universitaire de France (IUF), Paris, France. ✉e-mail: isabelle.bouyssou@outlook.com; pascal.campagne@pasteur.fr; dmenard@pasteur.fr; dmenard@unistra.fr

