## [Transparent Peer Review file · Communications Biology]

Genomic analysis of *Plasmodium vivax* field isolates circulating in sub-Saharan Africa

Corresponding Author: Professor Didier Ménard

Version 0:

Reviewer comments:

Reviewer #1

(Remarks to the Author)

I would like to begin by acknowledging the significant effort that went into this study. The genomic analysis of *Plasmodium vivax* field isolates from sub-Saharan Africa is a valuable contribution to understanding the genetic diversity of this parasite in a region where malaria remains a pressing public health challenge. The authors have made commendable progress in providing insights into the global genetic structure of *P. vivax* and have successfully identified regions of genomic differentiation that could be important for future research into drug resistance and parasite adaptation. Their use of modern sequencing technologies and analytical methods is commendable, and the study adds meaningful data to a growing body of work on *P. vivax* epidemiology.

I am glad to see whole-genome sequencing of 133 *P. vivax* isolates from 10 African countries (including Angola, Ethiopia, Madagascar, and Mauritania) was performed. This data was supplemented with 204 publicly available sequences from regions like Southeast Asia and South America, creating a dataset of 337 samples. The study used selective whole-genome amplification (sWGA) to address the challenge of low parasitemia in clinical samples.

Although the authors have produced a well-executed study, there is one core issue that limits the reliability of their conclusions. This article sets out to challenge the traditional understanding that *Plasmodium vivax* is limited to Duffy-positive individuals by analyzing the genetic diversity and potential adaptation mechanisms in Duffy-negative populations. However, despite the ambitious goal, the study faces significant limitations, particularly due to the insufficient collection of Duffy-negative samples. The majority of the samples used were Duffy-positive, and only a small subset (18 individuals) were homozygous Duffy-negative, from which meaningful genomic data could not be extracted due to low parasitemia. The conclusion drawn from the study is that there is no clear genetic signature indicating a specific adaptation of *P. vivax* to Duffy-negative hosts. However, this conclusion is undermined by the lack of robust data from Duffy-negative samples, which leaves the question of alternative invasion pathways largely unexplored. Without sufficient representation of Duffy-negative individuals, the association analysis between invasion-related genes and Duffy-negativity cannot be considered conclusive.

In the Global genetic diversity section of the results, the study primarily focuses on the geographic clustering of *P. vivax* populations. The clustering reflects geographic regions—such as East/West Africa, the Indian Ocean, and South America—and these distinctions are based on overall genetic diversity, not the Duffy status of the hosts. The clustering observed in this section relates to the global genetic structure of the parasite populations rather than any specific adaptation to Duffy-negativity. The genetic distinctions are linked to geographical locations and historical population migrations, not the interaction between *P. vivax* and the Duffy antigen. This part of the results is unrelated to Duffy status. It only sets the stage for understanding the broad genetic diversity of *P. vivax* across different regions.

In the Genomic islands of genetic differentiation section, the researchers tested whether the low frequency of Duffy antigens in sub-Saharan African populations could exert selective pressure on *P. vivax* parasites. This selective pressure might lead to genetic differentiation, particularly in regions of the genome that are involved in the parasite's ability to invade red blood cells through Duffy-independent pathways. However, the analysis is still largely based on Duffy-positive samples, since the Duffy-negative samples could not be fully sequenced due to low parasitemia. The "genomic selection stress test" they conducted essentially looks for genomic regions (called genomic islands of differentiation) that might be under selection pressure because of factors like Duffy antigen distribution, but the focus remains on Duffy-positive populations. The study did not find strong evidence of selective pressure related to Duffy-negativity in these genomic regions. Instead, the

differentiation signals were mostly linked to drug resistance genes rather than invasion-related genes like PvDBP or PvRBP.

The section on genetic diversity of genes associated with drug resistance seems tangential to the main focus of the paper, which is the potential adaptation of *P. vivax* to Duffy-negative individuals. While drug resistance is an important issue in malaria research and relevant in a whole-genome sequencing study, its inclusion here feels more like an obligatory mention due to the nature of the dataset rather than contributing directly to the paper's primary goal. The drug resistance analysis doesn't provide any insight into how *P. vivax* might be adapting to Duffy-negativity or invasion-related mechanisms, making it feel somewhat out of place in the context of the paper's main research question. This section may indeed be included to fill up the word count, or to cover a broader range of genomic findings, which is typical in comprehensive genomic reports, but its relevance to the paper's core objective is minimal.

For the GWAS part, since the majority of the data came from Duffy-positive individuals, it's not surprising that the GWAS didn't find anything significant. GWAS is a powerful tool, but it relies on genetic diversity and variation in the population being studied. In this case, the Duffy-positive samples used (doesn't matter if homozygous or heterozygous) may not have provided enough genetic variation related to invasion mechanisms. Without a meaningful comparison between Duffy-positive and Duffy-negative individuals, the chances of identifying any associations were low from the start.

I hereby suggest a major revision, and my suggestions are as follows:

To fully explore the potential adaptations of *Plasmodium vivax* to Duffy-negative individuals, I recommend to include a more balanced sample set. Ideally, the study should consist of at least 50% Duffy-negative samples and 50% Duffy-positive samples. This balanced approach would enable a more robust analysis and comparison of invasion-related genes like PvDBP and PvRBP. By having a larger and balanced sample set, the analysis could more effectively determine if these invasion-related genes form distinct clusters: One cluster representing Duffy-positive infections. Another cluster potentially representing Duffy-negative infections, indicating a possible adaptation. If clustering of these genes aligns with the Duffy status of the hosts, it would provide strong evidence that genetic adaptations in these genes are driving the parasite's ability to infect Duffy-negative individuals. If increasing the sample size is not feasible, the study should focus on population genetic research with the current data and avoid drawing conclusions about Duffy-negative host adaptations.

I also recommend including the FWS test in future analyses to better differentiate between monoinfections and mixed infections. Without a clear understanding of whether a sample represents a monoinfection or a mixed infection, the calculation of heterozygosity may be skewed. This could lead to inaccurate genetic differentiation metrics such as *Gst* or *Fst*, which may falsely suggest higher genetic diversity or differentiation than actually exists. The absence of FWS makes it difficult to confidently determine the true monoinfection status of a sample. This is particularly problematic in regions with high transmission rates, where mixed infections are more common. By incorporating the FWS test, you would ensure more accurate heterozygosity calculations, which would in turn improve the reliability of the genetic differentiation metrics.

Reviewer #2

(Remarks to the Author)

This substantial body of work describes the genomic analysis of 133 *P. vivax* field isolates from 10 different African countries (Angola, Burundi, Comoros, Djibouti, Egypt, Eritrea, Ethiopia, Madagascar, Mauritania, and Sudan) with 204 published sequences of *P. vivax* field isolates from eight other countries in Southeast Asia, the Pacific Coral Triangle countries and South American countries for a total dataset of 337 *vivax* samples from 18 countries. While many of the results may not be different to what has already been published in other studies, and the authors did not find associations between parasite genetic diversity and adaptation to Duffy-negative human hosts, the paper represents interesting analyses, is well written and thought provoking and a significant addition to the set of global *vivax* genomes. *Vivax* genomics is challenging as the authors say and the authors have done as well as they could.

I have a few suggestions that would improve the paper:

1. Please provide more details in the suppl methods of the GATK4 parameters used to call snps. Snp calling is still an art rather than a science and being able to reproduce the calls will be useful for readers.
2. Please add the citations for all of the 204 *P. vivax* published sequences in the Suppl Table 1. This will aid the reader and give credit to the hard work undertaken by the authors of the other studies.
3. Similarly, the authors should contextualize their findings with previous published studies on the population genetics of *vivax* globally, acknowledging and comparing more what has been done before them. This will help the reader understand how the results presented are novel or agree with previous studies. For example, how does the genetic structure presented in the Hupalo 2016 paper agree with the authors' results? And the loci identified in the same paper and others as under selection and diverged in different populations? Are they the same?
4. In the Data Availability section, the authors state: "Data availability All raw sequence data is available from the European Nucleotide Archive (www.ebi.ac.uk/ena; see Supplementary Data 1 for accession numbers). These data include samples from the LSHTM returning travelers (PRJEB44419) and the MalariaGEN *P. vivax* Genome Variation project (see ref. 10)." Can the authors clarify: were their samples collected as part of MalariaGen activities, or what is the relationship with MalariaGen? There was a recent paper by Siegal et al Nat Comms 2024 that described 615 Pv genomes, was there overlap between the samples analyzed in that study?

5. Can the authors state the range of years over which the samples were collected and provide in the text? Its important because of the changing epidemiology of malaria. I am wondering if any inferences can be made about the increase or decrease of Pv transmission in the sampled countries over the time of sampling, since this could affect the COI of infections.

Version 1:

Reviewer comments:

Reviewer #1

(Remarks to the Author)

The authors have made significant improvements to the manuscript in response to my previous comments. I appreciate their careful revisions, which have enhanced the clarity and accuracy of their findings. They have appropriately reframed their conclusions regarding Plasmodium vivax adaptation to Duffy-negative individuals, ensuring that their results are not overinterpreted. Additionally, they have clearly acknowledged the limitations imposed by the lack of high-quality Duffy-negative samples, making the study's scope and contributions more transparent. Overall, these revisions meet my expectations, and I commend the authors for their efforts in strengthening the manuscript.

In particular, the authors have revised multiple sections to clarify the limitations imposed by the lack of high-quality Duffy-negative samples. They have adjusted the abstract and discussion to explicitly state that their study does not provide conclusive evidence of P. vivax adaptation to Duffy-negative hosts due to low parasitemia and sequencing challenges. The results section now appropriately frames the genomic differentiation analysis as a general population genetics study, rather than an investigation of Duffy-independent invasion pathways. In the GWAS section, they have tempered their conclusions, acknowledging that the unbalanced sample set prevented meaningful associations. Additionally, they have included the FWS test to distinguish between monoclonal and polyclonal infections, ensuring more reliable heterozygosity and genetic differentiation estimates. These changes make the study more rigorous and prevent overinterpretation of its findings.

Despite these improvements, a fundamental issue remains: the lack of sufficient Duffy-negative samples prevents a definitive analysis of P. vivax adaptation to Duffy-negative individuals. Without a more balanced dataset—including a larger proportion of Duffy-negative samples with high-quality genomic data—it is impossible to determine whether specific genetic adaptations are facilitating invasion in these hosts. This limitation affects key analyses, including GWAS and selection tests, which were conducted almost exclusively on Duffy-positive samples. However, I recognize that obtaining such samples is extremely challenging due to low parasitemia and logistical constraints, and I appreciate that the authors have been transparent about this issue. Given the current limitations, I consider this study to be a well-executed population genetics analysis rather than an investigation into alternative invasion pathways.

Given these considerations, I accept the authors' explanation and understand that obtaining additional Duffy-negative samples is not feasible at this stage. While this limitation prevents the study from fully addressing its original aim, the authors have made substantial efforts to refine their conclusions and present their findings within the appropriate context. Their revisions ensure that the manuscript is as rigorous and accurate as possible given the available data. As a result, I agree to accept the manuscript, recognizing its value as a well-executed population genetics study that provides important insights into P. vivax diversity in Africa.

Reviewer #2

(Remarks to the Author)

The authors have adequately responded to my comments in their response letter. But the marked-up files that were provided do not correspond to the changes stated. For example: "Please read now (data availability section, page 16): 'It is important to clarify that while our study includes sequences from the publicly available MalariaGEN dataset, our samples were independently collected and were not generated as part of MalariaGEN activities.'" On pg 16 of the marked-up document there is no such text in the body of the manuscript.

Similarly, "Pages 11-12: 'Our findings align with previous observations reported by Hupalo et al.,²⁵ which..." there is no such text anywhere in the document, and the Hupalo paper is cited as Ref 31 not 25.

and several other changes requested are not seen.

Also: Please add the citations for all of the 204 P. vivax published sequences in the Suppl Table 1; not just in the Data availability section, as requested. You want to make it easier on the reader to find these datasets, not make them dig. Many thanks.

Version 2:

Reviewer comments:

Reviewer #2

(Remarks to the Author)

Thank you to the authors for addressing my concerns of not being able to clearly see the changes to the manuscript. All the changes have been made. This is going to be an excellent addition to the *P. vivax* genomic epidemiology literature.

Reviewers' comments:

Reviewer #1 (Remarks to the Author):

I would like to begin by acknowledging the significant effort that went into this study. The genomic analysis of *Plasmodium vivax* field isolates from sub-Saharan Africa is a valuable contribution to understanding the genetic diversity of this parasite in a region where malaria remains a pressing public health challenge. The authors have made commendable progress in providing insights into the global genetic structure of *P. vivax* and have successfully identified regions of genomic differentiation that could be important for future research into drug resistance and parasite adaptation. Their use of modern sequencing technologies and analytical methods is commendable, and the study adds meaningful data to a growing body of work on *P. vivax* epidemiology.

I am glad to see whole-genome sequencing of 133 *P. vivax* isolates from 10 African countries (including Angola, Ethiopia, Madagascar, and Mauritania) was performed. This data was supplemented with 204 publicly available sequences from regions like Southeast Asia and South America, creating a dataset of 337 samples. The study used selective whole-genome amplification (sWGA) to address the challenge of low parasitemia in clinical samples.

Reply: We thank the reviewer for appreciating our work.

Although the authors have produced a well-executed study, there is one core issue that limits the reliability of their conclusions. This article sets out to challenge the traditional understanding that *Plasmodium vivax* is limited to Duffy-positive individuals by analyzing the genetic diversity and potential adaptation mechanisms in Duffy-negative populations. However, despite the ambitious goal, the study faces significant limitations, particularly due to the insufficient collection of Duffy-negative samples. The majority of the samples used were Duffy-positive, and only a small subset (18 individuals) were homozygous Duffy-negative, from which meaningful genomic data could not be extracted due to low parasitemia. The conclusion drawn from the study is that there is no clear genetic signature indicating a specific adaptation of *P. vivax* to Duffy-negative hosts. However, this conclusion is undermined by the lack of robust data from Duffy-negative samples, which leaves the question of alternative invasion pathways largely unexplored. Without sufficient representation of Duffy-negative individuals, the association analysis between invasion-related genes and Duffy-negativity cannot be considered conclusive.

Reply: We fully agree with this comment. Our initial aim was to compare strains of *P. vivax* infecting Duffy-positive and Duffy-negative individuals. However, as the reviewer rightly points out, we were unable to obtain interpretable sequences from Duffy-negative patients due to the extremely low parasite densities in these samples. Overcoming this challenge remains difficult, as no reliable methods currently exist to concentrate parasites or enhance sequence quality in such cases. Additionally, logistical and financial constraints further limit the collection of new Duffy-negative samples.

Despite these limitations, we are confident that our study provides novel insights into the genetic diversity of *P. vivax* strains circulating in regions with a high prevalence of Duffy-negative individuals. Notably, parasites infecting Duffy-positive individuals may also have previously infected Duffy-negative hosts, indirectly reflecting selective pressure. Thus, we believe our genomic selection stress test remains relevant, as it identifies potential genomic regions under selection pressure, including those influenced by Duffy antigen distribution. We acknowledge that future studies with larger and more balanced sample sets, including Duffy-

negative individuals, will be essential to draw definitive conclusions regarding *P. vivax* adaptation to Duffy-negative hosts

In the Global genetic diversity section of the results, the study primarily focuses on the geographic clustering of *P. vivax* populations. The clustering reflects geographic regions—such as East/West Africa, the Indian Ocean, and South America—and these distinctions are based on overall genetic diversity, not the Duffy status of the hosts. The clustering observed in this section relates to the global genetic structure of the parasite populations rather than any specific adaptation to Duffy-negativity. The genetic distinctions are linked to geographical locations and historical population migrations, not the interaction between *P. vivax* and the Duffy antigen. This part of the results is unrelated to Duffy status. It only sets the stage for understanding the broad genetic diversity of *P. vivax* across different regions.

Reply: Yes, we agree with this comment. We confirm that this paragraph does not mention that clustering is related to the interaction between *P. vivax* and the Duffy antigen.

In the Genomic islands of genetic differentiation section, the researchers tested whether the low frequency of Duffy antigens in sub-Saharan African populations could exert selective pressure on *P. vivax* parasites. This selective pressure might lead to genetic differentiation, particularly in regions of the genome that are involved in the parasite's ability to invade red blood cells through Duffy-independent pathways. However, the analysis is still largely based on Duffy-positive samples, since the Duffy-negative samples could not be fully sequenced due to low parasitemia. The "genomic selection stress test" they conducted essentially looks for genomic regions (called genomic islands of differentiation) that might be under selection pressure because of factors like Duffy antigen distribution, but the focus remains on Duffy-positive populations. The study did not find strong evidence of selective pressure related to Duffy-negativity in these genomic regions. Instead, the differentiation signals were mostly linked to drug resistance genes rather than invasion-related genes like PvDBP or PvRBP.

Reply: We have clarified that the identified genomic islands of differentiation do not reflect selective pressures associated with Duffy negativity. An additional sentence has been added on page 7: *'Although no evidence was found for specific genetic adaptations to Duffy-independent invasion in the regions examined, these findings underscore the complexity of P. vivax evolution in regions where Duffy negativity predominates. Future studies incorporating larger datasets of high-quality genomes from Duffy-negative individuals will be essential to conclusively identify loci under selection for alternative invasion pathways.'* Moreover, the discussion has been revised to emphasize this distinction and prevent overinterpretation of the current findings

The section on genetic diversity of genes associated with drug resistance seems tangential to the main focus of the paper, which is the potential adaptation of *P. vivax* to Duffy-negative individuals. While drug resistance is an important issue in malaria research and relevant in a whole-genome sequencing study, its inclusion here feels more like an obligatory mention due to the nature of the dataset rather than contributing directly to the paper's primary goal. The drug resistance analysis doesn't provide any insight into how *P. vivax* might be adapting to Duffy-negativity or invasion-related mechanisms, making it feel somewhat out of place in the context of the paper's main research question. This section may indeed be included to fill up the word count, or to cover a broader range of genomic findings, which is typical in comprehensive genomic reports, but its relevance to the paper's core objective is minimal.

Reply: We agree with this comment; however, we believe it is valuable to present these data, as information on molecular signatures potentially associated with antimalarial drug

resistance in *P. vivax* parasites remains scarce in Africa. While the drug resistance section may seem tangential to the primary focus of Duffy-independent adaptation, it provides important insights into selective pressures acting on *P. vivax* populations in the region. To improve coherence, we have revised the text to better integrate this section within the broader context of our study.

For the GWAS part, since the majority of the data came from Duffy-positive individuals, it's not surprising that the GWAS didn't find anything significant. GWAS is a powerful tool, but it relies on genetic diversity and variation in the population being studied. In this case, the Duffy-positive samples used (doesn't matter if homozygous or heterozygous) may not have provided enough genetic variation related to invasion mechanisms. Without a meaningful comparison between Duffy-positive and Duffy-negative individuals, the chances of identifying any associations were low from the start.

Reply: We acknowledge the limitations of the GWAS analysis due to the unbalanced sample set, which predominantly consisted of Duffy-positive individuals. The text has been revised to better reflect this limitation and temper our conclusions. Specifically, we have added the following statement to the discussion section (pages 13): *'Despite increasing evidence of P. vivax infections in Duffy-negative individuals, our study was unable to identify clear genomic signatures of parasite adaptation. This limitation primarily stems from the low parasitemia observed in these individuals, which restricted the availability of high-quality genomic sequences. As a result, potential adaptive mechanisms may remain undetected, particularly if they involve complex or subtle evolutionary processes. Overcoming these challenges will require larger, well-balanced datasets, improved parasite enrichment techniques, and functional studies to elucidate alternative invasion pathways.'*

I hereby suggest a major revision, and my suggestions are as follows:

To fully explore the potential adaptations of Plasmodium vivax to Duffy-negative individuals, I recommend to include a more balanced sample set. Ideally, the study should consist of at least 50% Duffy-negative samples and 50% Duffy-positive samples. This balanced approach would enable a more robust analysis and comparison of invasion-related genes like PvDBP and PvRBP. By having a larger and balanced sample set, the analysis could more effectively determine if these invasion-related genes form distinct clusters: One cluster representing Duffy-positive infections. Another cluster potentially representing Duffy-negative infections, indicating a possible adaptation. If clustering of these genes aligns with the Duffy status of the hosts, it would provide strong evidence that genetic adaptations in these genes are driving the parasite's ability to infect Duffy-negative individuals. If increasing the sample size is not feasible, the study should focus on population genetic research with the current data and avoid drawing conclusions about Duffy-negative host adaptations.

Reply: We acknowledge the limitation due to the small number of Duffy-negative samples, which is primarily a result of the challenges in obtaining high-quality data from low-parasitemia infections. While increasing the sample size is not currently feasible, our study still provides important insights into *P. vivax* diversity in regions with a high prevalence of Duffy-negative individuals. Consequently, we have reframed our conclusions regarding Duffy-negativity as hypotheses for future investigations using larger datasets.

To address this, we have added the following statements:

Abstract, page 2. 'Despite the limited number of interpretable sequences from Duffy-negative individuals - attributable to low parasitemia - and the lack of clear evidence of selective

pressure acting on invasion-related genes of the *P. vivax* parasite populations circulating in sub-Saharan Africa, our study offers valuable insights into the genetic diversity of *P. vivax* and lays the groundwork for future research exploring parasite adaptation to Duffy-negative hosts.'

Page 4: 'While the limited availability of high-quality Duffy-negative sequences prevents definitive conclusions on parasite adaptation, this work lays the foundation for future studies with larger and more balanced datasets.'

Page 4: 'Despite our efforts, *P. vivax* genome sequences from 18 homozygous Duffy-negative patients could not be properly exploited due to insufficient parasitemia levels, which resulted in low DNA amplification and inadequate sequencing quality (see Methods for details). This limitation highlights the ongoing technical challenges in obtaining high-quality genomic data from Duffy-negative individuals, an issue that remains a bottleneck for understanding potential adaptations of *P. vivax* to these hosts.'

Page 7: 'Although no evidence was found for specific genetic adaptations to Duffy-independent invasion in the regions examined, these findings underscore the complexity of *P. vivax* evolution in regions where Duffy negativity predominates. Future studies incorporating larger datasets of high-quality genomes from Duffy-negative individuals will be essential to conclusively identify loci under selection for alternative invasion pathways.'

Page 10: 'Alternatively, adaptations to infect Duffy-negative individuals, if they exist, may involve molecular mechanisms too complex or subtle to be detected by broad-scale genomic association methods, particularly given the constraints imposed by the limited availability of high-quality Duffy-negative sequences.'

Page 13: 'Despite increasing evidence of *P. vivax* infections in Duffy-negative individuals, our study was unable to identify clear genomic signatures of parasite adaptation. This limitation primarily stems from the low parasitemia observed in these individuals, which restricted the availability of high-quality genomic sequences. As a result, potential adaptive mechanisms may remain undetected, particularly if they involve complex or subtle evolutionary processes. Overcoming these challenges will require larger, well-balanced datasets, improved parasite enrichment techniques, and functional studies to elucidate alternative invasion pathways.'

I also recommend including the FWS test in future analyses to better differentiate between mono-infections and mixed infections. Without a clear understanding of whether a sample represents a mono-infection or a mixed infection, the calculation of heterozygosity may be skewed. This could lead to inaccurate genetic differentiation metrics such as G_{st} or F_{st} , which may falsely suggest higher genetic diversity or differentiation than actually exists. The absence of FWS makes it difficult to confidently determine the true mono-infection status of a sample. This is particularly problematic in regions with high transmission rates, where mixed infections are more common. By incorporating the FWS test, you would ensure more accurate heterozygosity calculations, which would in turn improve the reliability of the genetic differentiation metrics.

Reply: We agree with the reviewer's comment. We performed the FWS test to distinguish mono-infections from mixed infections, ensuring more accurate heterozygosity calculations and differentiation metrics. This test strengthens the robustness of our genetic diversity and structure analyses.

To address this, we have added the following statements:

Page 4: 'To ensure a robust analysis, we assessed within-host diversity by calculating F_{ws} values to distinguish monoclonal infections from polyclonal infections and to estimate heterozygosity.'

Page 5: 'To assess the presence of polyclonal infections, we evaluated within-host diversity by calculating F_{ws} coefficients²⁴. Empirically, isolates with $F_{ws} < 0.95$ are generally considered as polyclonal, while those with $F_{ws} > 0.95$ are expected to be monoclonal. This analysis

revealed that approximately one-third of the isolates (32.4%) could be classified as polyclonal (Table S2 and Figure S2).’

Page 15: ‘Analysis of genotypic data. We applied the *F_{ws}* test to distinguish between monoinfections and polyclonal infections. Given the potential bias introduced by mixed infections (particularly in analyses assessing genetic diversity, population structure, and genotype-phenotype associations), we adopted a dual approach to ensure the robustness and transparency of our findings. Unless specified otherwise, all data analyses were done with R 4.3.0 (R Core Team 2023). Within-host diversity index was calculated across loci as $F_{ws} = 1 - H_w/H_s$, where H_w is the within-host heterozygosity and H_s is the expected heterozygosity at population level²⁴. For the main analyses, including the genome-wide association study (GWAS), population structure inference using STRUCTURE, principal coordinate analysis (PCoA), and Tajima’s D calculation, we excluded polyclonal infections. This decision was made to avoid confounding effects arising from multiple parasite genotypes within a single sample, which can inflate measures of heterozygosity and obscure the detection of population structure and genetic associations. By focusing solely on monoinfections, we ensured that each sample represented a single parasite genome, providing more accurate estimates of genetic variation and population differentiation. To assess the impact of excluding polyclonal infections on our findings, we conducted complementary analyses that included both monoclonal and polyclonal infections. These additional analyses were consistent with the main results and are presented in the Supplementary Information (Figure S3). This approach ensures that our conclusions are robust while maintaining transparency regarding the potential influence of polyclonal infections on genetic diversity and population structure.’

Figure S2 and table S2 presenting the *F_{ws}* test results have been included

Reviewer #2 (Remarks to the Author):

This substantial body of work describes the genomic analysis of 133 *P. vivax* field isolates from 10 different African countries (Angola, Burundi, Comoros, Djibouti, Egypt, Eritrea, Ethiopia, Madagascar, Mauritania, and Sudan) with 204 published sequences of *P. vivax* field isolates from eight other countries in Southeast Asia, the Pacific Coral Triangle countries and South American countries for a total dataset of 337 *vivax* samples from 18 countries. While many of the results may not be different to what has already been published in other studies, and the authors did not find associations between parasite genetic diversity and adaptation to Duffy-negative human hosts, the paper represents interesting analyses, is well written and thought provoking and a significant addition to the set of global *vivax* genomes. *Vivax* genomics is challenging as the authors say and the authors have done as well as they could.

Reply: We thank the reviewer for appreciating our work.

I have a few suggestions that would improve the paper:

1. Please provide more details in the suppl methods of the GATK4 parameters used to call snps. Snp calling is still an art rather than a science and being able to reproduce the calls will be useful for readers.

Reply: We appreciate the reviewer’s comment regarding the need for detailed information on the GATK4 parameters used for SNP calling. We have now added comprehensive details in the supplementary methods to ensure the reproducibility of our results. This includes specific thresholds for quality filtering, genotype calling settings, and other relevant parameters used during the analysis.

To address this, we have added the following statements:

Page 5: 'To ensure robust variant calling, we applied hard filtering based on several summary statistics. Variants were excluded if they met any of the following criteria: QD < 2.0, QUAL < 30.0, SOR > 3.0, FS > 60.0, MQ < 40.0, MQRankSum < -12.5, ReadPosRankSum < -8.0.

Further details of these filters and an overview of the workflow are provided in **Figure S1.**'
Supplementary Information, Supplementary methods: Variant calling

2. Please add the citations for all of the 204 *P. vivax* published sequences in the Suppl Table 1. This will aid the reader and give credit to the hard work undertaken by the authors of the other studies.

Reply: This information is now provided. All references for the 204 previously published *P. vivax* sequences have been included, ensuring proper credit to the original studies. Data availability section now includes references for all published sequences.

3. Similarly, the authors should contextualize their findings with previous published studies on the population genetics of *vivax* globally, acknowledging and comparing more what has been done before them. This will help the reader understand how the results presented are novel or agree with previous studies. For example, how does the genetic structure presented in the Hupalo 2016 paper agree with the authors' results? And the loci identified in the same paper and others as under selection and diverged in different populations? Are they the same?

Reply: We have expanded the discussion to compare our findings with key studies, including Hupalo et al. (2016) and Benavente et al. (2021). This contextualization highlights how our results align with or differ from previous work on *P. vivax* population genetics. We have added additional sentences in the discussion section:

Pages 11-12: 'Our findings align with previous observations reported by Hupalo et al.,²⁵ which demonstrated a clear geographic genetic structure of *P. vivax* populations worldwide, including low levels of admixture. Similarly, our analysis revealed well-defined clusters for African, Asian, and South American populations, with limited gene flow between regions. In addition, the work of Benavente et al.²⁴ identified loci under selective pressure, particularly in drug resistance genes (*pvkelch10*, *pvmrp1*), which differ from the loci identified in our study (*pvmr1*, *pvdhfr*, *pvdhps*). This discrepancy likely reflects regional variations in drug use and the selective pressures acting on *P. vivax* populations in Africa compared to other regions.'

4. In the Data Availability section, the authors state: "Data availability All raw sequence data is available from the European Nucleotide Archive (www.ebi.ac.uk/ena; see Supplementary Data 1 for accession numbers). These data include samples from the LSHTM returning travelers (PRJEB44419) and the MalariaGEN *P. vivax* Genome Variation project (see ref. 10)." Can the authors clarify: were their samples collected as part of MalariaGen activities, or what is the relationship with MalariaGen? There was a recent paper by Siegal et al Nat Comms 2024 that described 615 Pv genomes, was there overlap between the samples analyzed in that study?

Reply: We clarified the relationship between our study and MalariaGEN, noting that our samples were not collected as part of MalariaGEN activities. Please read now (data availability section, page 16): 'It is important to clarify that while our study includes sequences from the publicly available MalariaGEN dataset, our samples were independently collected and were not generated as part of MalariaGEN activities.'

5. Can the authors state the range of years over which the samples were collected and provided in the text? Its important because of the changing epidemiology of malaria. I am

wondering if any inferences can be made about the increase or decrease of Pv transmission in the sampled countries over the time of sampling, since this could affect the COI of infections.

Reply: We provided the range of years during which samples were collected. We have added the following statements:

Page 4: 'Samples. Dried blood spots and whole blood samples were collected between 2016-2021 from symptomatic *P. vivax* patients at health centers in Ethiopia (N = 66), Madagascar (N = 27), Mauritania (N = 2), and Angola (N = 12). Additionally, we included DNA extracts from samples obtained from symptomatic travelers returning to France from the Comoros (N = 8), Mauritania (N = 5), Djibouti (N = 3), Sudan (N = 3), Madagascar (N = 2), Eritrea (N = 2), Ethiopia (N = 1), Burundi (N = 1), and Egypt (N = 1).'

Page 13: 'Biological samples. All biological samples used in this study were obtained from *P. vivax*-infected blood samples after informed consent was obtained between 2016 and 2021.'

Reviewers' comments:

Response to Reviewer #1:

The authors have made significant improvements to the manuscript in response to my previous comments. I appreciate their careful revisions, which have enhanced the clarity and accuracy of their findings. They have appropriately reframed their conclusions regarding *Plasmodium vivax* adaptation to Duffy-negative individuals, ensuring that their results are not overinterpreted. Additionally, they have clearly acknowledged the limitations imposed by the lack of high-quality Duffy-negative samples, making the study's scope and contributions more transparent. Overall, these revisions meet my expectations, and I commend the authors for their efforts in strengthening the manuscript.

In particular, the authors have revised multiple sections to clarify the limitations imposed by the lack of high-quality Duffy-negative samples. They have adjusted the abstract and discussion to explicitly state that their study does not provide conclusive evidence of *P. vivax* adaptation to Duffy-negative hosts due to low parasitemia and sequencing challenges. The results section now appropriately frames the genomic differentiation analysis as a general population genetics study, rather than an investigation of Duffy-independent invasion pathways. In the GWAS section, they have tempered their conclusions, acknowledging that the unbalanced sample set prevented meaningful associations. Additionally, they have included the FWS test to distinguish between monoclonal and polyclonal infections, ensuring more reliable heterozygosity and genetic differentiation estimates. These changes make the study more rigorous and prevent overinterpretation of its findings.

Despite these improvements, a fundamental issue remains: the lack of sufficient Duffy-negative samples prevents a definitive analysis of *P. vivax* adaptation to Duffy-negative individuals. Without a more balanced dataset—including a larger proportion of Duffy-negative samples with high-quality genomic data—it is impossible to determine whether specific genetic adaptations are facilitating invasion in these hosts. This limitation affects key analyses, including GWAS and selection tests, which were conducted almost exclusively on Duffy-positive samples. However, I recognize that obtaining such samples is extremely challenging due to low parasitemia and logistical constraints, and I appreciate that the authors have been transparent about this issue. Given the current limitations, I consider this study to be a well-executed population genetics analysis rather than an investigation into alternative invasion pathways.

Given these considerations, I accept the authors' explanation and understand that obtaining additional Duffy-negative samples is not feasible at this stage. While this limitation prevents the study from fully addressing its original aim, the authors have made substantial efforts to refine their conclusions and present their findings within the appropriate context. Their revisions ensure that the manuscript is as rigorous and accurate as possible given the available data. As a result, I agree to accept the manuscript, recognizing its value as a well-executed population genetics study that provides important insights into *P. vivax* diversity in Africa.

Reply: We sincerely thank the reviewer for their positive and constructive evaluation of our revised manuscript. We greatly appreciate their thoughtful and detailed comments, which helped us to substantially improve the clarity, rigor, and contextual framing of our findings. We fully acknowledge, as the reviewer highlighted, the current limitations due to the difficulty of obtaining high-quality genomic data from Duffy-negative individuals. We also agree that

our study should be positioned as a contribution to the understanding of *Plasmodium vivax* population genetics in Africa, rather than as a conclusive investigation into alternative invasion pathways.

We are pleased that the reviewer considers our revised manuscript acceptable and recognizes the value of our work. We believe that our findings provide an important foundation for future studies that will require larger and better-balanced sample sets to further explore *P. vivax* adaptation mechanisms.

Thank you again for your careful review and support.

Update of the previous responses to Reviewer #2 :

In response to the reviewers' comments, we have carefully updated the manuscript. All the modifications that were made in response to the reviewers' requests are now clearly indicated in the version of the manuscript and the supplemental material with tracked changes (All marks) **in red font**.

We hope that these updates will facilitate the review of our revised submission.

We sincerely thank the reviewers and the editor for their constructive feedback, which has helped us to significantly improve the clarity and rigor of the manuscript.

This substantial body of work describes the genomic analysis of 133 *P. vivax* field isolates from 10 different African countries (Angola, Burundi, Comoros, Djibouti, Egypt, Eritrea, Ethiopia, Madagascar, Mauritania, and Sudan) with 204 published sequences of *P. vivax* field isolates from eight other countries in Southeast Asia, the Pacific Coral Triangle countries and South American countries for a total dataset of 337 *vivax* samples from 18 countries. While many of the results may not be different to what has already been published in other studies, and the authors did not find associations between parasite genetic diversity and adaptation to Duffy-negative human hosts, the paper represents interesting analyses, is well written and thought provoking and a significant addition to the set of global *vivax* genomes. *Vivax* genomics is challenging as the authors say and the authors have done as well as they could.

Reply: We thank the reviewer for appreciating our work.

I have a few suggestions that would improve the paper:

1. Please provide more details in the suppl methods of the GATK4 parameters used to call snps. Snp calling is still an art rather than a science and being able to reproduce the calls will be useful for readers.

Reply: We appreciate the reviewer's comment regarding the need for detailed information on the GATK4 parameters used for SNP calling. We have now added comprehensive details in the supplementary methods to ensure the reproducibility of our results. This includes specific thresholds for quality filtering, genotype calling settings, and other relevant parameters used during the analysis.

To address this, we have added the following statements:

Page 7, line 8: 'To ensure robust variant calling, we applied hard filtering based on several summary statistics. Variants were excluded if they met any of the following criteria: QD < 2.0, QUAL < 30.0, SOR > 3.0, FS > 60.0, MQ < 40.0, MQRankSum < -12.5, ReadPosRankSum < -8.0. Further details of these filters and an overview of the workflow are provided in **Figure S1**.'

Supplementary Information, Supplementary methods: Variant calling (page 6)

2. Please add the citations for all of the 204 *P. vivax* published sequences in the Suppl Table 1. This will aid the reader and give credit to the hard work undertaken by the authors of the other studies.

Reply: This information is now provided. All references for the 204 previously published *P. vivax* sequences have been included, ensuring proper credit to the original studies. Data availability section now includes references for all published sequences (see pages 13-18, Supplementary Material)

3. Similarly, the authors should contextualize their findings with previous published studies on the population genetics of *vivax* globally, acknowledging and comparing more what has been done before them. This will help the reader understand how the results presented are novel or agree with previous studies. For example, how does the genetic structure presented in the Hupalo 2016 paper agree with the authors' results? And the loci identified in the same paper and others as under selection and diverged in different populations? Are they the same?

Reply: We have expanded the discussion to compare our findings with key studies, including Hupalo et al. (2016) and Benavente et al. (2021). This contextualization highlights how our results align with or differ from previous work on *P. vivax* population genetics. We have added additional sentences in the discussion section:

Start Page 14 and Pages 18-19: 'Our findings align with previous observations reported by Hupalo et al.,²⁵ which demonstrated a clear geographic genetic structure of *P. vivax* populations worldwide, including low levels of admixture. Similarly, our analysis revealed well-defined clusters for African, Asian, and South American populations, with limited gene flow between regions. In addition, the work of Benavente et al.²⁴ identified loci under selective pressure, particularly in drug resistance genes (*pvkelch10*, *pvmrp1*), which differ from the loci identified in our study (*pvm-dr1*, *pvdhfr*, *pvdhps*). This discrepancy likely reflects regional variations in drug use and the selective pressures acting on *P. vivax* populations in Africa compared to other regions.'

4. In the Data Availability section, the authors state: "Data availability All raw sequence data is available from the European Nucleotide Archive (www.ebi.ac.uk/ena; see Supplementary Data 1 for accession numbers). These data include samples from the LSHTM returning travelers (PRJEB44419) and the MalariaGEN *P. vivax* Genome Variation project (see ref. 10)." Can the authors clarify: were their samples collected as part of MalariaGen activities, or what is the relationship with MalariaGen? There was a recent paper by Siegal et al Nat Comms 2024 that described 615 Pv genomes, was there overlap between the samples analyzed in that study?

Reply: We clarified the relationship between our study and MalariaGEN, noting that our samples were not collected as part of MalariaGEN activities. Please read now (data availability section, page 23): 'It is important to clarify that while our study includes sequences from the publicly available MalariaGEN dataset, our samples were independently collected and were not generated as part of MalariaGEN activities.'

5. Can the authors state the range of years over which the samples were collected and provided in the text? Its important because of the changing epidemiology of malaria. I am wondering if any inferences can be made about the increase or decrease of Pv transmission in the sampled countries over the time of sampling, since this could affect the COI of infections.

Reply: We provided the range of years during which samples were collected. We have added the following statements:

Page 6: 'Samples. Dried blood spots and whole blood samples were collected between 2016-2021 from symptomatic *P. vivax* patients at health centers in Ethiopia (N = 66), Madagascar (N = 27), Mauritania (N = 2), and Angola (N = 12). Additionally, we included DNA extracts from samples obtained from symptomatic travelers returning to France from the Comoros (N = 8), Mauritania (N = 5), Djibouti (N = 3), Sudan (N = 3), Madagascar (N = 2), Eritrea (N = 2), Ethiopia (N = 1), Burundi (N = 1), and Egypt (N = 1).'

Page 21: 'Biological samples. All biological samples used in this study were obtained from *P. vivax*-infected blood samples after informed consent was obtained between 2016 and 2021.'

Additional Response to Reviewer #2 :

The authors have adequately responded to my comments in their response letter. But the marked-up files that were provided do not correspond to the changes stated. For example: "Please read now (data availability section, page 16): 'It is important to clarify that while our study includes sequences from the publicly available MalariaGEN dataset, our samples were independently collected and were not generated as part of MalariaGEN activities.' On pg 16 of the marked-up document there is no such text in the body of the manuscript.

Similarly, "Pages 11-12: 'Our findings align with previous observations reported by Hupalo et al.,²⁵ which..." there is no such text anywhere in the document, and the Hupalo paper is cited as Ref 31 not 25.

Reply: We confirm that the sentence referring to Hupalo et al. was already present in the revised manuscript, and that the reference corresponds to Ref 26 in the final version.

and several other changes requested are not seen.

Reply: See reply below (Update of the previous responses to Reviewer #2).

Also: Please add the citations for all of the 204 *P. vivax* published sequences in the Suppl Table 1; not just in the Data availability section, as requested. You want to make it easier on the reader to find these datasets, not make them dig. Many thanks.

Reply: We thank the reviewer for their thorough reading and helpful comments.

We confirm that:

- The clarification regarding the use of MalariaGEN data has been added to the Data Availability section (page 23, tracked change version, all marks).
- The reference to Hupalo et al. and the description of global population structure have been included in the Discussion section (page 14 & 18-19, tracked change version, all marks).
- The Supplementary Table S1 has been updated with a new "Reference" column listing the source publications for each public genome sequence used (page 13-18, tracked change version, all marks).

All these modifications are clearly marked in the attached version of the manuscript with tracked changes to facilitate review.

We sincerely appreciate the reviewer's feedback, which helped us improve the clarity and completeness of the manuscript.